# Sample Complexity of Automated Mechanism Design

**Maria-Florina Balcan, Tuomas Sandholm, Ellen Vitercik**
School of Computer Science
Carnegie Mellon University
Pittsburgh, PA 15213
{ninamf,sandholm,vitercik}@cs.cmu.edu

## Abstract

The design of revenue-maximizing combinatorial auctions, i.e. multi-item auctions over bundles of goods, is one of the most fundamental problems in computational economics, unsolved even for two bidders and two items for sale. In the traditional economic models, it is assumed that the bidders' valuations are drawn from an underlying distribution and that the auction designer has perfect knowledge of this distribution. Despite this strong and oftentimes unrealistic assumption, it is remarkable that the revenue-maximizing combinatorial auction remains unknown. In recent years, *automated mechanism design* has emerged as one of the most practical and promising approaches to designing high-revenue combinatorial auctions. The most scalable automated mechanism design algorithms take as input *samples* from the bidders' valuation distribution and then search for a high-revenue auction in a rich auction class. In this work, we provide the first sample complexity analysis for the standard hierarchy of deterministic combinatorial auction classes used in automated mechanism design. In particular, we provide tight sample complexity bounds on the number of samples needed to guarantee that the empirical revenue of the designed mechanism on the samples is close to its expected revenue on the underlying, unknown distribution over bidder valuations, for each of the auction classes in the hierarchy. In addition to helping set automated mechanism design on firm foundations, our results also push the boundaries of learning theory. In particular, the hypothesis functions used in our contexts are defined through multi-stage combinatorial optimization procedures, rather than simple decision boundaries, as are common in machine learning.

## 1 Introduction

Multi-item, multi-bidder auctions have been studied extensively in economics, operations research, and computer science. In a *combinatorial auction (CA)*, the bidders may submit bids on bundles of goods, rather than on individual items alone, and thereby they may fully express their complex valuation functions. Notably, these functions may be non-additive due to the presence of complementary or substitutable goods for sale. There are many important and practical applications of CAs, ranging from the US government's wireless spectrum license auctions to sourcing auctions, through which companies coordinate the procurement and distribution of equipment, materials and supplies.

One of the most important and tantalizing open questions in computational economics is the design of *optimal auctions*, that is, auctions that maximize the seller's expected. In the standard economic model, it is assumed that the bidders' valuations are drawn from an underlying distribution and that the mechanism designer has perfect information about this distribution. Astonishingly, even with this strong assumption, the optimal CA design problem is unsolved even for auctions with just two distinct items for sale and two bidders. A monumental advance in the study of optimal auction design was the characterization of the optimal 1-item auction [Myerson, 1981]. However, the problem becomes

significantly more challenging with multiple items for sale. In particular, Conitzer and Sandholm proved that the problem of finding a revenue-maximizing deterministic CA is NP-complete [Conitzer and Sandholm, 2004]. We note here that it is well-known that randomization can increase revenue in CAs, but we focus on deterministic CAs in this work because in many applications, randomization is not palatable and very few, if any, randomized CAs are used in practice.

In recent years, a novel approach known as *automated mechanism design* (AMD) has been adopted to attack the revenue-maximizing auction design problem [Conitzer and Sandholm, 2002, Sandholm, 2003]. In the most scalable strand of AMD, algorithms have been developed which take samples from the bidders' valuation distributions as input, optimize over a rich class of auctions, and return an auction which is high-performing over the sample [Likhodedov and Sandholm, 2004, 2005, Sandholm and Likhodedov, 2015]. AMD algorithms have yielded deterministic mechanisms with the highest known revenues in the contexts used for empirical evaluations [Sandholm and Likhodedov, 2015]. This approach relaxes the unrealistic assumption that the mechanism designer has perfect information about the bidders' valuation distribution.

However, until now, there was no formal characterization of the number of samples required to guarantee that the empirical revenue of the designed mechanism on the samples is close to its expected revenue on the underlying, unknown distribution over bidder valuations. In this paper, we provide that missing link. We present tight sample complexity guarantees over an extensive hierarchy of expressive CA families. These are the most commonly used auction families in AMD. The classes in the hierarchy are based on the classic VCG mechanism, which is a generalization of the well-known second-price, or Vickrey, single-item auction. The auctions we consider achieve significantly higher revenue than the VCG baseline by weighting bidders (multiplicatively increasing all of their bids) and boosting outcomes (additively increasing the liklihood that a particular outcome will be the result of the auction).

A major strength of our results is their applicability to any algorithm that determines the optimal auction over the sample, a nearly optimal approximation, or any other black box procedure. Therefore, they apply to any automated mechanism design algorithm, optimal or not. One of the key challenges in deriving these general sample complexity bounds is that to do so, we must develop deep insights into how changes to the auction parameters (the bidder weights and allocation boosts) effect the outcome of the auction (who wins which items and how much each bidder pays) and thereby the revenue of the auction. In our context, we show that the functions which determine the outcome of an auction are highly complex, consisting of multi-stage optimization procedures.

Therefore, the function classes we consider are much more challenging than those commonly found in machine learning contexts. Typically, for well-understood classes of functions used in machine learning, such as linear separators or other smooth curves in Euclidean spaces, there is a simple mapping from the parameters of a specific hypothesis to its prediction on a given example and a close connection between the distance in the parameter space between two parameter vectors and the distance in function space between their associated hypotheses. Roughly speaking, it is necessary to understand this connection in order to determine how many significantly different hypotheses there are over the full range of parameters. In our context, due to the inherent complexity of the classes we consider, connecting the parameter space to the space of revenue functions requires a much more delicate analysis. The key technical part of our work involves understanding this connection from a learning theoretic perspective. For the more general classes in the hierarchy, we use Rademacher complexity to derive our bounds, and for the auction classes with more combinatorial structure, we exploit that structure to prove pseudo-dimension bounds. This work is both of practical importance since we fill a fundamental gap in AMD, and of learning theoretical interest, as our sample complexity analysis requires a deep understanding of the structure of the revenue function classes we consider.

**Related Work.** In prior research, the sample complexity of revenue maximization has been studied primarily in the single-item or the more general single-dimensional settings [Elkind, 2007, Cole and Roughgarden, 2014, Huang et al., 2015, Medina and Mohri, 2014, Morgenstern and Roughgarden, 2015, Roughgarden and Schrijvers, 2016, Devanur et al., 2016], as well as some multi-dimensional settings which are reducible to the single-bidder setting [Morgenstern and Roughgarden, 2016]. In contrast, the combinatorial settings that we study are much more complex since the revenue functions consist of multi-stage optimization procedures that cannot be reduced to a single-bidder setting. The complexity intrinsic to the multi-item setting is explored in [Dughmi et al., 2014], who show that

for a single unit-demand bidder, when the bidder's values for the items may be correlated, $\Omega(2^m)$ samples are required to determine a constant-factor approximation to the optimal auction.

Learning theory tools such as pseudo-dimension and Rademacher complexity were used to prove strong guarantees in [Medina and Mohri, 2014, Morgenstern and Roughgarden, 2015, 2016], which analyze piecewise linear revenue functions and show that few samples are needed to learn over the revenue function classes in question. In a similar direction, bounds on the sample complexity of welfare-optimal item pricings have been developed [Feldman et al., 2015, Hsu et al., 2016]. Earlier work of Balcan et al. [2008] addressed sample complexity results for revenue maximization in unrestricted supply settings. In that context, the revenue function decomposes additively among bidders and does not apply to our combinatorial setting.

Despite the inherent complexity of designing high-revenue CAs, Morgenstern and Roughgarden use linear separability as a tool to prove that certain simple classes of multi-parameter auctions have small sample complexity. The auctions they study are sequential auctions with item and grand bundle pricings, as well as second-price item auctions with item reserve prices [Morgenstern and Roughgarden, 2016]. In the item pricing auctions, the bidders show up one at a time and the seller offers each item that remains at some price. Each buyer then chooses the subset of goods that maximizes her utility. In the grand bundle pricing auctions, the bidders are each offered the grand bundle in some fixed order, and the first bidder to have a value greater than the price buys it. They show that bounding the sample complexity of these sequential auctions can be reduced to the single-buyer setting.

In contrast, the auctions we study are more versatile than item pricing auctions, as they give the mechanism designer many more degrees of freedom than the number of items. This level of expressiveness allows the designer to increase competition between bidders, much like Myerson's optimal auction, and thus boost revenue. It is easy to construct examples where even simple AMAs achieve significantly greater revenue than sequential auctions with item and grand bundle prices. Moreover, even the simpler auction classes we consider pose a unique challenge because the parameters defining the auctions influence the multi-stage allocation procedure and resulting revenue in non-intuitive ways. This is unlike item and grand bundle pricing auctions, as well as second-price item auctions, which are simple by design. Our function classes therefore require us to understand the specific form of the weighted VCG payment rule and its interaction with the parameter space. Thus, our context and techniques diverge from those in [Morgenstern and Roughgarden, 2016].

Finally, there is a wealth of work on characterizing the optimal CA for restricted settings and designing mechanisms which achieve high, if not optimal revenue in specific contexts. Due to space constraints, in Section A of the supplementary materials, we describe these results as well as what is known theoretically about the classes in the hierarchy of deterministic CAs we study.

## 2  Preliminaries, notation, and the combinatorial auction hierarchy

In the following section, we explain the basic mechanism design problem, fix notation, and then describe the hierarchy of combinatorial auction families we study.

**Mechanism Design Preliminaries.** We consider the problem of selling $m$ heterogeneous goods to $n$ bidders. This means that there are $2^m$ different bundles of goods, $B = \{b_1, \ldots, b_{2^m}\}$. Each bidder $i \in [n]$ is associated with a set-wise valuation function over the bundles, $v_i : B \to \mathbb{R}$. We assume that the bidders' valuations are drawn from a distribution $\mathcal{D}$.

Every auction is defined by an *allocation function* and a *payment function*. The allocation function determines which bidders receive which items based on their bids and the payment function determines how much the bidders need to pay based on their bids and the allocation. It is up to the mechanism designer to determine which allocation and payment functions should be used. In our context, the two functions are fixed based on the samples from $\mathcal{D}$ before the bidders submit their bids.

Each auction family that we consider has a design based on the classic *Vickrey-Clarke-Groves mechanism (VCG)*. The VCG mechanism, which we describe below, is the canonical *strategy-proof* mechanism, which means that every bidder's dominant strategy is to bid truthfully. In other words, for every Bidder $i$, no matter the bids made by the other bidders, Bidder $i$ maximizes her expected utility (her value for her allocation minus the price she pays) by bidding her true value. Therefore, we describe the VCG mechanism assuming that the bids equal the bidders' true valuations.

The VCG mechanism allocates the items such that the social welfare of the bidders, that is, the sum of each bidder's value for the items she wins, is maximized. Intuitively, each winning bidder then pays her bid minus a "rebate" equal to the increase in welfare attributable to Bidder $i$'s presence in the auction. This form of the payment function is crucial to ensuring that the auction is strategy-proof. More concretely, the allocation of the VCG mechanism is the disjoint set of subsets $(b_1^*, \ldots, b_n^*) \subseteq B$ that maximizes $\sum v_i(b_i^*)$. Meanwhile, let $(b_1^{-i}, \ldots, b_n^{-i})$ be the disjoint set of subsets that maximizes $\sum_{j \neq i} v_j(b_j^{-i})$. Then Bidder $i$ must pay $\sum_{j \neq i} \left[ v_j(b_j^{-i}) - v_j(b_j^*) \right] = v_i(b_i^*) - \left[ \sum v_j(b_j^*) - \sum_{j \neq i} v_j(b_j^{-i}) \right]$. In the special case where there is one item for sale, the VCG mechanism is known as the second price, or Vickrey, auction, where the highest bidder wins the item and pays the second highest bid. We note that every auction in the classes we study is strategy-proof, so we may assume that the bids equal the bidders' valuations.

**Notation.** We study auctions with $n$ bidders and $m$ items. We refer to the bundle of all $m$ items as the *grand bundle*. In total, there are $(n+1)^m$ possible allocations, which we denote as the vectors $\mathcal{O} = \{\vec{o}_1, \ldots, \vec{o}_{(n+1)^m}\}$. Each allocation vector $\vec{o}_i$ can be written as $(o_{i,1}, \ldots, o_{i,n})$, where $o_{i,j} = b_\ell \in B$ denotes the bundle of items allocated to Bidder $j$ in allocation $\vec{o}_i$. We use the notation $\vec{v}_1 = (v_1(b_1), \ldots, v_1(b_{2^m}))$ and $\vec{v} = (\vec{v}_1, \ldots, \vec{v}_n)$ to denote a vector of bidder valuation functions. We say that $rev_A(\vec{v})$ is the revenue of an auction $A$ on the valuation vector $\vec{v}$. Denoting the payment of any one bidder under auction $A$ given valuation vector $\vec{v}$ as $p_{i,A}(\vec{v})$, we have that $rev_A(\vec{v}) = \sum_{i=1}^n p_{i,A}(\vec{v})$. Finally, $U$ is an upper bound on the revenue achievable for any auction over the support of the bidders' valuation distribution.

**Auction Classes.** We now give formal definitions of the CA families in the hierarchy we study. See Figure 1 for the hierarchical organization of the auction classes, together with the papers which introduced each family.

*Affine maximizer auctions (AMAs).* An AMA $A$ is defined by a set of weights per bidder $(w_1, \ldots, w_n) \subset \mathbb{R}_{>0}$ and boosts per allocation $(\lambda(\vec{o}_1), \ldots, \lambda(\vec{o}_{(n+1)^m})) \subset \mathbb{R}$. An auction $A$ uniquely corresponds to a set of these parameters, so we write $A = (w_1, \ldots, w_n, \lambda(\vec{o}_1), \ldots, \lambda(\vec{o}_{(n+1)^m}))$. To simplify notation, we write $\lambda_i = \lambda(\vec{o}_i)$ interchangeably. These parameters allow the mechanism designer to multiplicatively boost any bidder's bids by their corresponding weight and to increase the likelihood that any one allocation is returned as the output of an auction. More concretely, the allocation $\vec{o}^*$ of an AMA $A$ is the one which maximizes the weighted social welfare, i.e. $\vec{o}^* = \text{argmax}_{\vec{o}_i \in \mathcal{O}} \left\{ \sum_{j=1}^n w_j v_j(o_{i,j}) + \lambda(\vec{o}_i) \right\}$. The payment function of $A$ has the same form as the VCG payment rule, with the parameters factored in to ensure that the auction remains strategy-proof. In particular, for all $j \in [n]$, the payments are $p_{j,A}(\vec{v}) = \frac{1}{w_j} \left[ \sum_{\ell \neq j} w_\ell v_\ell(o_{-j,\ell}) + \lambda(\vec{o}_{-j}) - \sum_{\ell \neq j} w_\ell v_\ell(o_\ell^*) - \lambda(\vec{o}^*) \right]$, where $\vec{o}_{-j} = \text{argmax}_{\vec{o}_i \in \mathcal{O}} \left\{ \sum_{\ell \neq j} w_\ell v_\ell(o_{i,\ell}) + \lambda(\vec{o}_i) \right\}$.

We assume that $H_{\underline{w}} \leq w_i \leq H_{\overline{w}}$, $\lambda_i \leq H_\lambda$, and $v_i(b_\ell) \leq H_v$ for some $H_{\underline{w}}, H_{\overline{w}}, H_\lambda, H_v \in \mathbb{R}_{\geq 0}$. It is typical to assume an upper bound (here, $H_v$) on the bidders' valuation for any bundle. This is related to the fact that an upper bound on a target function's range is always assumed in standard machine learning sample complexity bounds. Intuitively, generalizability depends on how much any one sample can skew the empirical average of a hypothesis, or in this case, auction. The bounds on the AMA parameters are closely related to the bound on the bidders' valuations $H_v$. For example, it is a simple exercise to see that we need not search for a lambda value which is greater than $H_v$.

*Virtual valuation combinatorial auctions (VVCAs).* VVCAs are a subset of AMAs. The defining characteristic of a VVCA is that each $\lambda(\vec{o}_j)$ is split into $n$ terms such that $\lambda(\vec{o}_j) = \sum_{i=1}^n \lambda_i(\vec{o}_j)$ where $\lambda_i(\vec{o}_j) = c_{i,b}$ for all allocations $\vec{o}_j$ that give Bidder $i$ exactly bundle $b \in B$.

*λ-auctions.* λ-auctions are the subclass of AMAs where $w_i = 1$ for all $i \in [n]$.

*Mixed bundling auctions (MBAs).* The class of MBAs is parameterized by a constant $c \geq 0$ which can be seen as a discount for any bidder who receives the grand bundle. Formally, the $c$-MBA is the λ-auction with $\lambda(\vec{o}) = c$ if some bidder receives the grand bundle in allocation $\vec{o}$ and 0 otherwise.

*Mixed bundling auctions with reserve prices (MBARPs).* MBARPs are identical to MBAs though with *reserve prices*. In a single-item VCG auction (i.e. second price auction) with a reserve price, the

item is only sold if the highest bidder's bid exceeds the reserve price, and the winner must pay the maximum of the second highest bid and the reserve price. We describe how this intuition generalizes to MBAs in Section 3.

**Generalization bounds.** In order to derive sample complexity bounds which apply to any algorithm that determines the optimal auction over the sample, a nearly optimal approximation, or any other black-box procedure, we derive uniform convergence sample complexity bounds with respect to the auction classes we examine. Formally, we define the sample complexity of uniform convergence over an auction class $\mathcal{A}$ as follows.

**Definition 1** (Sample complexity of uniform convergence over $\mathcal{A}$). *We say that $N(\epsilon, \delta, \mathcal{A})$ is the sample complexity of uniform convergence over $\mathcal{A}$ if for any $\epsilon, \delta \in (0, 1)$, if $\mathcal{S} = \left\{\vec{v}^1, \dots, \vec{v}^N\right\}$ is a sample of size $N \geq N(\epsilon, \delta, \mathcal{A})$ drawn at random from $\mathcal{D}$, with probability at least $1 - \delta$, for all auctions $A \in \mathcal{A}$, $\left| \frac{1}{N} \sum_{i=1}^{N} rev_A\left(\vec{v}^i\right) - \mathbb{E}_{\vec{v} \sim \mathcal{D}}\left[rev_A(\vec{v})\right] \right| \leq \epsilon.$*

# 3  Sample complexity bounds over the hierarchy of auction classes

In this section, we provide an overview of our sample complexity guarantees over the hierarchy of auction classes we consider (Section 3.1 and 3.2). We show that more structured classes require drastically fewer samples to learn over. We conclude with a note about sample complexity guarantees for algorithms that find an approximately optimal mechanism over a sample, as opposed to the optimal mechanism. All omitted proofs are presented in full in the supplementary material.

## 3.1  The sample complexity of AMA, VVCA, and $\lambda$-auction revenue maximization

We begin by analyzing the most general families in the CA hierarchy — AMAs, VVCAs, and $\lambda$-auctions — proving a general upper bound and class-specific lower bounds.

**Theorem 1.** *The sample complexity of uniform convergence over the classes of $n$-bidder, $m$-item AMAs, VVCAs, and $\lambda$-Auctions is $N = \widetilde{O}\left(\left[Un^m\sqrt{m}\left(U + n^{m/2}\right)/\epsilon\right]^2\right)$. Moreover, for $\lambda$-Auctions, $N = \Omega\left(n^m\right)$ and for VVCAs, $N = \Omega\left(2^m\right)$.*

We derive the upper bound by analyzing the Rademacher complexity of the class of $n$-bidder, $m$-item AMA revenue functions. For a family of functions $G$ and a finite sample $S = \{x_1, \dots, x_N\}$ of size $N$, the empirical Rademacher complexity is defined as $\widehat{\mathcal{R}}_S(G) = \mathbb{E}_\sigma[\sup_{g \in G} \frac{1}{N} \sum \sigma_i g(x_i)]$, where $\sigma = (\sigma_1, \dots, \sigma_N)$, with $\sigma_i$s independent uniform random variables taking values in $\{-1, 1\}$. The Rademacher complexity of $G$ is defined as $\mathcal{R}_N(G) = \mathbb{E}_{S \sim \mathcal{D}^N}[\widehat{\mathcal{R}}_S(G)]$.

The AMA revenue function, defined in Section 2, can be summarized as a multi-stage optimization procedure: determine the weighted-optimal allocation and then compute the $n$ different payments, each of which requires a separate optimization procedure. Luckily, we are able to decompose the revenue functions into small components, each of which is easier to analyze on its own, and then combine our results to prove the following theorem about this class of revenue functions as a whole.

**Theorem 2.** *Let $\mathcal{F}$ be the set of $n$-bidder, $m$-item AMA revenue functions $rev_A$ such that $A = \left(w_1, \dots, w_n, \lambda_1, \dots, \lambda_{(n+1)^m}\right), H_{\underline{w}} \leq |w_i| \leq H_{\overline{w}}, |\lambda_i| \leq H_\lambda$. Then*

$$\mathcal{R}_N(\mathcal{F}) = O\left(\frac{n^{m+2}\left(H_{\overline{w}}H_v + H_\lambda\right)}{H_{\underline{w}}}\sqrt{\frac{m\log n}{N}}\left(\frac{n\hat{H}_v\left(nH_{\overline{w}} + H_\lambda\right)}{H_{\underline{w}}} + \sqrt{n^m \log N}\right)\right),$$

*where $\hat{H}_v = \max\{H_v, 1\}$.*

*Proof sketch.* First, we describe how we split each revenue function into smaller, easier to analyze atoms, which together allow us to bound the Rademacher complexity of the class of AMA revenue functions. To this end, it is well-known (e.g. [Mohri et al., 2012]) that if every function $f$ in a class $\mathcal{F}$ can be written as the summation of two functions $g$ and $h$ from classes $\mathcal{G}$ and $\mathcal{H}$, respectively, then $\mathcal{R}_N(\mathcal{F}) \leq \mathcal{R}_N(\mathcal{G}) + \mathcal{R}_N(\mathcal{H})$. Therefore, we split each revenue function into $n + 1$ components such that the sum of these components equals the revenue function.

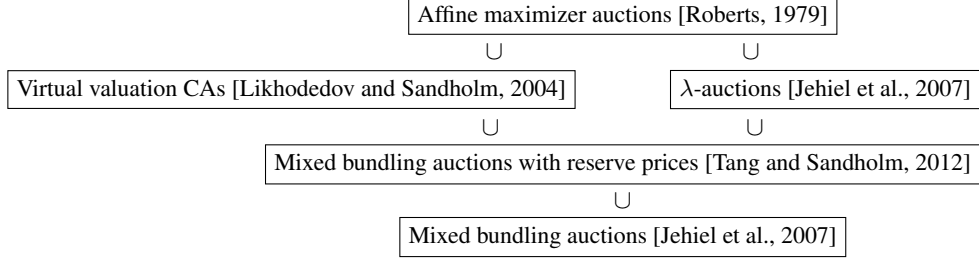

Figure 1: The hierarchy of deterministic CA families. Generality increases upward in the hierarchy.

With this objective in mind, let $\vec{o}_A^*(\vec{v})$ be the outcome of the AMA $A$ on the bidding instance $\vec{v}$, i.e. $\vec{o}_A^* = \text{argmax}_{\vec{o}_i \in \mathcal{O}} \left\{ \sum_{j=1}^n w_j v_j (o_{i,j}) + \lambda_i \right\}$ and let $\phi_{A,-j}(\vec{v})$ be the weighted social welfare of the welfare-maximizing outcome without Bidder $j$'s participation. In other words, $\phi_{A,-j}(\vec{v}) = \max_{\vec{o}_i \in \mathcal{O}} \left\{ \sum_{\ell \neq j} w_\ell v_\ell (o_{i,\ell}) + \lambda_i \right\}$. Then we can write

$$rev_A(\vec{v}) = \sum_{j=1}^n \frac{1}{w_j} \phi_{A,-j}(\vec{v}) - \sum_{i=1}^{(n+1)^m} \left( \sum_{j=1}^n \frac{1}{w_j} \sum_{\ell \neq j} w_\ell v_\ell(o_{i,\ell}) + \lambda_i \right) \mathbf{1}_{\vec{o}_i = \vec{o}_A^*(\vec{v})}.$$

We can now split $rev_A$ into $n+1$ simpler functions: $rev_{A,j}(\vec{v}) = \frac{1}{w_j} \phi_{A,-j}(\vec{v})$ for $j \in [n]$ and

$$rev_{A,n+1}(\vec{v}) = - \sum_{i=1}^{(n+1)^m} \left( \sum_{j=1}^n \frac{1}{w_j} \sum_{\ell \neq j} w_\ell v_\ell (o_{i,\ell}) + \lambda_i \right) \mathbf{1}_{\vec{o}_i = \vec{o}_A^*(\vec{v})},$$

so $rev_A(\vec{v}) = \sum_{j=1}^{n+1} rev_{A,j}(\vec{v})$. Intuitively, for $j \in [n]$, $rev_{A,j}$ is a weighted version of what the social welfare would be if Bidder $j$ had not participated in the auction, whereas $rev_{A,n+1}(\vec{v})$ measures the amount of revenue subtracted to ensure that the resulting auction is strategy-proof.

As to be expected, bounding the Rademacher complexity of each smaller class of functions $\mathcal{L}_j = \left\{ rev_{A,j} \mid (w_1, \ldots, w_n, \lambda_1, \ldots, \lambda_{(n+1)^m}), H_{\underline{w}} \leq |w_i| \leq H_{\overline{w}}, |\lambda_i| \leq H_\lambda \right\}$ for $j \in [n+1]$ is simpler than bounding the Rademacher complexity the class of revenue functions itself, and if $\mathcal{F}$ is the set of all $n$-bidder, $m$-item AMA revenue functions, then $\mathcal{R}_N(\mathcal{F}) \leq \sum_{j=1}^{n+1} \mathcal{R}_N(\mathcal{L}_j)$. In Lemma 2 and Lemma 3 of Section B.1 in the supplementary materials, we obtain bounds on $\mathcal{R}_N(\mathcal{L}_j)$ for $j \in [n+1]$ which lead us to our bound on $\mathcal{R}_N(\mathcal{F})$. $\qquad\square$

## 3.2 The sample complexity of MBA revenue maximization

Fortunately, these negative sample complexity results are not the end of the story. We do achieve polynomial sample complexity upper bounds for the important classes of mixed bundling auctions (MBAs) and mixed bundling auctions with reserve prices (MBARPs). We derive these sample complexity bounds by analyzing the pseudo-dimensions of these classes of auctions. In this section, we present our results in increasing complexity, beginning with the class of $n$-bidder, $m$-item MBAs, which we show has a pseudo-dimension of 2. We build on the proof of this result to show that the class of $n$-bidder, $m$-item MBARPs has a pseudo-dimension of $O\left(m^3 \log n\right)$.

We note that when we analyze the class of MBARPs, we assume additive reserve prices, rather than bundle reserve prices. In other words, each item has its own reserve price, and the reserve price of a bundle is the sum of its components' reserve prices, as opposed to each bundle having its own reserve price. We have good reason to make this restriction; in Section C.1, we prove that an exponential number of samples are required to learn over the class of MBARPs with bundle reserve prices.

Before we prove our sample complexity results, we fix some notation. For any $c$-MBA, let $rev_c(\vec{v})$ be its revenue on $\vec{v}$, which is determined in the exact same way as the general AMA revenue function with the $\lambda$ terms set as described in Section 2. Similarly, let $rev_{\vec{v}^\ell}(c)$ be the revenue of the $c$-MBA on $\vec{v}^\ell$ *as a function of c*. We will use the following result regarding the structure of $rev_c(\vec{v})$ in order to derive our pseudo-dimension results. The proof is in Section C of the supplementary materials.

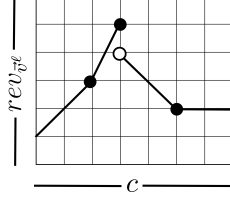

Figure 2: Example of $rev_{\vec{v}^\ell}(c)$.

**Lemma 1.** *There exists $c^* \in [0, \infty)$ such that $rev_{\vec{v}}(c)$ is non-decreasing on the interval $[0, c^*]$ and non-increasing on the interval $(c^*, \infty)$.*

The form of $rev_{\vec{v}}(c)$ as described in Lemma 1 is depicted in Figure 2. The full proof of the following pseudo-dimension bound can be found in Section C of the supplementary materials.

**Theorem 3.** *The pseudo-dimension of the class of $n$-bidder, $m$-item MBAs is 2.*

*Proof sketch.* First, we recall what we must show in order to prove that the pseudo-dimension of this class is 2 (for more on pseudo-dimension, see, for example, [Mohri et al., 2012]). The proof structure is similar to those involved in VC dimension derivations. To begin with, we must provide a set of two valuation vectors $\mathcal{S} = \{\vec{v}^1, \vec{v}^2\}$ that can be *shattered* by the class of MBA revenue functions. This means that there exist two *targets* $z^1, z^2 \in \mathbb{R}$ with the property that for any $T \subseteq \mathcal{S}$, there exists a $c_T \in C$ such that if $\vec{v}^i \in T$, then $rev_{c_T}(\vec{v}^i) \leq z^i$ and if $\vec{v}^i \notin T$, then $rev_{c_T}(\vec{v}^i) > z^i$. In other words, $\mathcal{S}$ can be labeled in every possible way by MBA revenue functions (whether or not $rev_c(\vec{v}^j)$ is greater than its target $z^j$). We must also prove that no set of three valuation vectors is shatterable.

Our construction of the set $\mathcal{S} = \{\vec{v}^1, \vec{v}^2\}$ that can be shattered by the set of MBAs can be found in the full proof of this theorem in Section C of the supplementary materials. We now show that no set of size $N \geq 3$ can be shattered by the class of MBAs. Fix one sample $\vec{v}^i \in \mathcal{S}$ and consider $rev_{\vec{v}^i}(c)$. From Lemma 1, we know that there exists $c_i^* \in [0, \infty)$, such that $rev_{\vec{v}^i}(c)$ is non-decreasing on the interval $[0, c_i^*]$ and non-increasing on the interval $(c_i^*, \infty)$. Therefore, there exist two thresholds $t_i^1 \in [0, c_i^*]$ and $t_i^2 \in (c_i^*, \infty) \cup \{\infty\}$ such that $rev_{\vec{v}^i}(c)$ is below its threshold for $c \in [0, t_i^1)$, above its threshold for $c \in (t_i^1, t_i^2)$, and below its threshold for $c \in (t_i^2, \infty)$. Now, merge these thresholds for all $N$ samples on the real line and consider the interval $(t_1, t_2)$ between two adjacent thresholds. The binary labeling of the samples in $\mathcal{S}$ on this interval is fixed. In other words, for any sample $\vec{v}^j \in \mathcal{S}$, $rev_{\vec{v}^j}(c)$ is either at least $z^j$ or strictly less than $z^j$ for all $c \in (t_1, t_2)$. There are at most $2N + 1$ intervals between adjacent thresholds, so at most $2N + 1$ different binary labelings of $\mathcal{S}$. Since we assumed $\mathcal{S}$ is shatterable, it must be that $2^N \leq 2N + 1$, so $N \leq 2$. $\square$

This result allows us to prove the following sample complexity guarantee.

**Theorem 4.** *The sample complexity of uniform convergence over the class of $n$-bidder, $m$-item MBAs is $N = O\left((U/\epsilon)^2 (\log(U/\epsilon) + \log(1/\delta))\right)$.*

**Mixed bundling auctions with reserve prices (MBARPs).** MBARPs are a variation on MBAs, with the addition of reserve prices. Reserve prices in the single-item case, as described in Section 2, can be generalized to the multi-item case as follows. We enlarge the set of agents to include the seller, who is now Bidder 0 and whose valuation for a set of items is the set's reserve price. Working in this expanded set of agents, the bidder weights are all 1 and the $\lambda$ terms are the same as in the standard MBA setup. Importantly, the seller makes no payments, no matter her allocation. More formally, given a vector of valuation functions $\vec{v}$, the MBARP allocation is $\vec{o}^* = \text{argmax}_{\vec{o} \in \mathcal{O}} \sum_{i=0}^n v_i(o_i) + \lambda(\vec{o})$. For each $i \in \{1, \ldots, n\}$, Bidder $i$'s payment is

$$p_{A,i}(\vec{v}) = \sum_{j \in \{0,\ldots,n\} \setminus \{i\}} v_j(o_{-i,j}) + \lambda(\vec{o}_{-i}) - \sum_{j \in \{0,\ldots,n\} \setminus \{i\}} v_j(o_j^*) - \lambda(\vec{o}^*),$$

where

$$\vec{o}_{-i} = \text{argmax}_{\vec{o} \in \mathcal{O}} \sum_{j \in \{0,\ldots,n\} \setminus \{i\}} v_j(o_j) + \lambda(\vec{o}).$$

As mentioned, we restrict our attention to item-specific reserve prices. In this case, the the reserve price of a bundle is the sum of the reserve prices of the items in the bundle.

Each MBARP is therefore parameterized by $m + 1$ values $(c, r_1, \ldots, r_m)$, where $r_i$ is the reserve price for the $i^{th}$ good. For a fixed valuation function vector $\vec{v} = (v_1(b_1), \ldots, v_1(b_{2^m}), \ldots, v_n(b_1), \ldots, v_n(b_{2^m}))$, we can analyze the MBARP revenue function on $\vec{v}$ as a mapping $rev_{\vec{v}} : \mathbb{R}^{m+1} \to \mathbb{R}$, where $rev_{\vec{v}}(c, r_1, \ldots, r_m)$ is the revenue of the MBARP parameterized by $(c, r_1, \ldots, r_m)$ on $\vec{v}$.

**Theorem 5.** *The psuedo-dimension of the class of $n$-bidder, $m$-item MBARPs with item-specific reserve prices is $O\left(m^3 \log n\right)$.*

*Proof sketch.* Let $\mathcal{S} = \left\{\vec{v}^1, \ldots, \vec{v}^N\right\}$ of size $N$ be a set of $n$-bidder valuation function samples that can be shattered by a set $C$ of $2^N$ MBARPs. This means that there exist $N$ targets $z^1, \ldots, z^N$ such that each MBARP in $C$ induces a binary labeling of the samples $\vec{v}^j$ in $\mathcal{S}$ (whether the revenue of the MBARP on $\vec{v}^j$ is greater than or less than $z^j$). Since $\mathcal{S}$ is shatterable, we can thus label $\mathcal{S}$ in every possible way using MBARPs in $C$.

This proof is similar to the proof of Theorem 3, where we split the real line into a set of intervals $\mathcal{I}$ such that for any $I \in \mathcal{I}$, the binary labeling of $\mathcal{S}$ by the $c$-MBA revenue function was fixed for all $c \in I$. In the case of MBARPs, however, the domain is $\mathbb{R}^{m+1}$, so we cannot split the domain into intervals in the same way. Instead, we show that we can split the domain into cells such that the binary labeling of $\mathcal{S}$ by the MBARP revenue function is a fixed linear function as we range over parameters in a single cell. In this way, we show that $N = O\left(m^3 \log n\right)$. $\square$

This is enough to prove the following guarantee.

**Theorem 6.** *The sample complexity of uniform convergence over the class of $n$-bidder, $m$-item MBARPs with item-specific reserve prices is $N = O\left((U/\epsilon)^2 \left(m^3 \log n \log(U/\epsilon) + \log(1/\delta)\right)\right)$.*

### 3.3 Sample complexity bounds for approximation algorithms

It may not always be computationally feasible to solve for the best auction over $\mathcal{S}$ for the given auction family. Rather, we may only be able to determine an auction $A$ that has average revenue over $\mathcal{S}$ that is within a $(1 + \alpha)$ multiplicative factor of the revenue-maximizing auction over $\mathcal{S}$ within the family. Nonetheless, in Theorem 11 of the supplementary materials, we prove that with slightly more samples, we can ensure that the expected revenue of $A$ is close to being with a $(1 + \alpha)$ multiplicative factor of the expected revenue of the optimal auction within the family with respect to the real distribution $\mathcal{D}$. We prove a similar bound for an additive factor approximation as well.

## 4 Conclusion

In this paper, we proved strong bounds on the sample complexity of uniform convergence for the well-studied and standard auction families that constitute the hierarchy of deterministic combinatorial auctions. We thereby answered a crucial question in the study of (automated) mechanism design: how to relate the performance of the mechanisms in the search space over the input samples to their expectation over the underlying—unknown—distribution. Specifically, for a fixed class of auctions, we determine the sample complexity necessary to ensure that with high probability, for any auction in that class, the average revenue over the sample is close to the expected revenue with respect to the underlying, unknown distribution over bidders' valuations. Our bounds apply to any algorithm that finds an optimal or approximately optimal auction over an input sample, and therefore to any automated mechanism design algorithm. Moreover, our results and analyses are of interest from a learning theoretic perspective because the function classes which make up the hierarchy of deterministic combinatorial auctions diverge significantly from well-understood hypothesis classes typically found in machine learning.

**Acknowledgments.** This work was supported in part by NSF grants CCF-1535967, CCF-1451177, CCF-1422910, IIS-1618714, IIS-1617590, IIS-1320620, IIS-1546752, ARO award W911NF-16-1-0061, a Sloan Research Fellowship, a Microsoft Research Faculty Fellowship, an NSF Graduate Research Fellowship, and a Microsoft Research Women's Fellowship.

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
