[Supplementary Material]

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

# A    Additional Related Work

In this section, we discuss additional related work regarding multi-dimensional auction design. The simplicity of Myerson's optimal *single-item* auction might lead one to hope that the optimal multi-item auction could be so elegantly characterizable [Myerson, 1981]. Recent work has made considerable progress toward this end (e.g. [Alaei et al., 2013, Bhalgat et al., 2013, Bhattacharya et al., 2010, Cai et al., 2012b,a, 2013, Daskalakis et al., 2014, Kleinberg and Weinberg, 2012]) but there is still relatively little known about optimal multi-item auction design. The problem has also garnered significant interest from a more applied perspective, resulting in significant advances from the artificial intelligence and machine learning communities (e.g. [Parkes and Ungar, 2000, Lahaie, 2011, Wurman and Wellman, 2000, Parkes et al., 2004, Amin et al., 2013, Mohri and Munoz, 2014, 2015]).

Revenue-maximizing mechanism design complements an active research area in theoretical computer science which strives to answer the question: can *simple* mechanisms achieve near-optimal revenue? This question was posed by Hartline and Roughgarden [2009], who left the precise definition of a simple mechanism open for interpretation. Recently, Morgenstern and Roughgarden [2015, 2016] proposed an auction class's pseudo-dimension as a formal means of defining simplicity. In particular, Morgenstern and Roughgarden [2016] complemented pseudo-dimension bounds with known approximation guarantees for the corresponding simple auction classes. See Morgenstern and Roughgarden [2016] and references therein for descriptions of these guarantees.

In contrast, most analyses of the revenue achieved by the classes that make up the hierarchy of deterministic CAs have been empirical [Sandholm, 2003, Likhodedov and Sandholm, 2004, 2005, Tang and Sandholm, 2012, Sandholm and Likhodedov, 2015]. However, from a theoretical standpoint, Roberts, when introducing the class of AMAs [Roberts, 1979], proved that they are the only *ex post* strategy-proof mechanisms over unrestricted domains of valuations[1]. Lavi et al. [2003] went on to prove that under certain natural assumptions, every incentive compatible CA is almost[2] an affine maximizer.

# B    Proofs from Section 3.1

## B.1    AMA Upper Bound Proofs

*Proof of Theorem 1.*  This bound follows from Theorem 2 and standard Rademacher complexity bounds (e.g. [Mohri et al., 2012]). The lower bounds are proved in Theorems 8 and 9   □

*Proof of Theorem 2.*  First, we describe how we split each revenue function into smaller, easier to analyze atoms, which together allow us to bound the Rademacher complexity of the class of AMA revenue functions. To this end, it is well-known (e.g. [Mohri et al., 2012]) that if every function $f$ in a class $\mathcal{F}$ can be written as the summation of two functions $g$ and $h$ from classes $\mathcal{G}$ and $\mathcal{H}$, respectively, then $\mathcal{R}_N(\mathcal{F}) \leq \mathcal{R}_N(\mathcal{G}) + \mathcal{R}_N(\mathcal{H})$. Therefore, we split each revenue function into $n+1$ components such that the sum of these components equals the revenue function.

With this objective in mind, let $\vec{o}_A^*(\vec{v})$ be the outcome of the AMA $A$ on the bidding instance $\vec{v}$, i.e. $\vec{o}_A^* = \mathrm{argmax}_{\vec{o}_i \in \mathcal{O}} \left\{ \sum_{j=1}^n w_j v_j\left(o_{i,j}\right) + \lambda_i \right\}$ and let $\phi_{A,-j}(\vec{v})$ be the weighted social welfare of the welfare-maximizing outcome without Bidder $j$'s participation. In other words, $\phi_{A,-j}(\vec{v}) = \max_{\vec{o}_i \in \mathcal{O}} \left\{ \sum_{\ell \neq j} w_\ell v_\ell\left(o_{i,\ell}\right) + \lambda_i \right\}$. Then we can write

$$rev_A(\vec{v}) = \sum_{j=1}^n \frac{1}{w_j} \phi_{A,-j}(\vec{v}) - \sum_{i=1}^{(n+1)^m} \left( \sum_{j=1}^n \frac{1}{w_j} \sum_{\ell \neq j} w_\ell v_\ell(o_{i,\ell}) + \lambda_i \right) \mathbf{1}_{\vec{o}_i = \vec{o}_A^*(\vec{v})}.$$

We can now split $rev_A$ into $n+1$ simpler functions: $rev_{A,j}(\vec{v}) = \frac{1}{w_j}\phi_{A,-j}(\vec{v})$ for $j \in [n]$ and

$$rev_{A,n+1}(\vec{v}) = -\sum_{i=1}^{(n+1)^m}\left(\sum_{j=1}^{n}\frac{1}{w_j}\sum_{\ell \neq j}w_\ell v_\ell\left(o_{i,\ell}\right) + \lambda_i\right)\mathbf{1}_{\vec{o}_i = \vec{o}_A^*(\vec{v})},$$

so $rev_A(\vec{v}) = \sum_{j=1}^{n+1} rev_{A,j}(\vec{v})$. Intuitively, for $j \in [n]$, $rev_{A,j}$ is a weighted version of what the social welfare would be if Bidder $j$ had not participated in the auction, whereas $rev_{A,n+1}(\vec{v})$ measures the amount of revenue subtracted to ensure that the resulting auction is strategy-proof.

As to be expected, bounding the Rademacher complexity of each smaller class of functions $\mathcal{L}_j = \left\{rev_{A,j} \mid (w_1,\ldots,w_n,\lambda_1,\ldots,\lambda_{(n+1)^m}), H_{\underline{w}} \leq |w_i| \leq H_{\overline{w}}, |\lambda_i| \leq H_\lambda\right\}$ for $j \in [n+1]$ is simpler than bounding the Rademacher complexity the class of revenue functions itself, and if $\mathcal{F}$ is the set of all $n$-bidder, $m$-item AMA revenue functions, then $\mathcal{R}_N(\mathcal{F}) \leq \sum_{j=1}^{n+1}\mathcal{R}_N(\mathcal{L}_j)$. In Lemma 2 and Lemma 3 of Section B.1 in the supplementary materials, we obtain bounds on $\mathcal{R}_N(\mathcal{L}_j)$ for $j \in [n+1]$. In fact, we have that $\sum_{j=1}^{n}\mathcal{R}_N(\mathcal{L}_j) = O\left(\mathcal{R}_N(\mathcal{L}_{n+1})\right)$. Therefore,

$$\mathcal{R}_N(\mathcal{F}) \leq O\left(\mathcal{R}_N(\mathcal{L}_j)\right)$$
$$= O\left(\frac{n^{m+2}\left(H_{\overline{w}}H_v + H_\lambda\right)}{H_{\underline{w}}}\sqrt{\frac{m\log n}{N}}\left(\frac{n\hat{H}_v\left(nH_{\overline{w}} + H_\lambda\right)}{H_{\underline{w}}} + \sqrt{n^m\log N}\right)\right),$$

as claimed. $\qquad\square$

**Lemma 2.** *For $j \in [n]$,*

$$\mathcal{R}_N(\mathcal{L}_j) = O\left(\frac{n^m\hat{H}_v(nH_{\overline{w}} + H_\lambda)}{H_{\underline{w}}}\sqrt{\frac{m\log n}{N}}\right).$$

*Proof.* At a high level, we will show that every function $\phi_{A,-j}$, which is the weighted version of what the social welfare would be under the AMA $A$ if Bidder $j$ had not participated in the auction, can be written as the maximum over $O(n^m)$ linear functions. Each linear function maps from the high dimensional space $\mathbb{R}^{n2^m+1}$, but its corresponding weight vector has a small $\ell_1$ norm, which is a good sign in Rademacher complexity analyses, as we describe below. We take advantage of these structural properties in our Rademacher analysis, as follows.

First, we fix the notation that we will need for this proof. Let $\Phi_j$ be the set of functions $\phi_{A,-j}$ for all AMAs $A$. In other words, $\Phi_j = \{\phi_{A,-j} \mid A = (w_1,\ldots,w_n,\lambda_1,\ldots,\lambda_{(n+1)^m}), H_{\underline{w}} \leq |w_i| \leq H_{\overline{w}}, |\lambda_i| \leq H_\lambda\}$. Recall that the vector $\vec{v}$ has the form and $\vec{v} = (\vec{v}_1,\ldots,\vec{v}_n)$ where $\vec{v}_i = (v_i(b_1),\ldots,v_i(b_{2^m}))$ for some ordering of all $2^m$ bundles $b_1,\ldots,b_{2^m}$. In this proof, we will make use of the function $\sigma : [\#(\text{allocations})] \times [\#(\text{bidders})] \rightarrow [\#(\text{bundles})]$, where $\sigma(i,t)$ is the index of the bundle that Bidder $t$ receives in allocation $\vec{o}_i$. For example, if Bidder 7 receives the bundle $b_2$ in outcome $\vec{o}_i$, then $\sigma(i,7) = 2$.

Recall that $\phi_{A,-j}(\vec{v}) = \max_{\vec{o}_i}\left\{\sum_{\ell \neq j}w_\ell v_\ell(o_{i,\ell}) + \lambda_i\right\}$. We will now show that for every allocation $\vec{o}_i$, we can write $\sum_{\ell \neq j}w_\ell v_\ell(o_{i,\ell}) + \lambda_i$ as a linear function. In particular, $\sum_{\ell \neq j}w_\ell v_\ell(o_{i,\ell}) + \lambda_i$ is a linear function $h^i_{A,j}$ from $\mathbb{R}^{n2^m+1}$ to $\mathbb{R}$, since we can write $h^i_{A,j}(\vec{v},1) = (\vec{v},1) \cdot \vec{a}^i_{A,j}$, where

$$\vec{a}^i_{A,j}[\ell] = \begin{cases} w_t & \text{if } \ell = 2^m(t-1) + \sigma(i,t) \text{ and } t \neq j \\ \lambda_i & \text{if } \ell = n2^m + 1 \\ 0 & \text{otherwise,} \end{cases}.$$

To see why $\vec{a}^i_{A,j}$ has this form, notice that if $\ell = 2^m(t-1) + \sigma(i,t)$, then $\ell$ is the index in $\vec{v}$ of Bidder $t$'s valuation for bundle it receives in allocation $\vec{o}_i$. In other words, $\vec{v}[\ell] = v_t(o_{i,t})$. We set $\vec{a}^i_{A,j}[\ell]$ to $w_t$ so that when we dot $\vec{a}^i_{A,j}$ with $(\vec{v},1)$, we obtain the summand $w_t v_t(o_{i,t})$. We repeat this process for all $t \neq j$. If we also set the final index of $\vec{a}^i_{A,j}$ to $\lambda_i$, i.e. $\vec{a}^i_{A,j}[n2^m + 1] = \lambda_i$, then $(\vec{v},1) \cdot \vec{a}^i_{A,j} = \sum_{\ell \neq j}w_\ell v_\ell(o_{i,\ell}) + \lambda_i$.

Notice that each vector $\vec{a}^i_{A,j}$ is high-dimensional but very sparse. In fact, its $\ell_1$-norm is relatively small: $\|\vec{a}^i_{A,j}\|_1 \leq nH_{\overline{w}} + H_\lambda$. Moreover, the set of all "input" vectors $(\vec{v},1)$ has a bounded $\ell_\infty$

norm: $||(\vec{v}, 1)||_\infty \leq \max\{H_v, 1\} = \hat{H}_v$. Therefore, we may use special bounds for sets of linear functions with small $\ell_1$-norm. Specifically, it is well known that if $\mathcal{K}$ is the set of linear functions in $\mathbb{R}^d$ of the form $\mathcal{K} = \{\vec{x} \mapsto \vec{w} \cdot \vec{x} \mid ||\vec{w}||_1 \leq B\}$ and if with probability one, $||\vec{x}||_\infty \leq X_\infty$, then $\mathcal{R}_N(\mathcal{K}) \leq X_\infty B \sqrt{2 \log d / N}$. Therefore, if $\mathcal{H}_j^i$ is the class of linear functions $h_{A,j}^i(\vec{v}) = \vec{a}_{A,j}^i \cdot (\vec{v}, 1)$ for all AMAs $A$, i.e.

$$\mathcal{H}_j^i = \left\{ h_{A,j}^i \mid A = (w_1, \ldots, w_n, \lambda_1, \ldots, \lambda_{(n+1)^m}), H_{\underline{w}} \leq |w_i| \leq H_{\overline{w}}, |\lambda_i| \leq H_\lambda \right\},$$

then

$$\mathcal{R}_N(\mathcal{H}_j^i) \leq \hat{H}_v(nH_{\overline{w}} + H_\lambda)\sqrt{\frac{2\log(n2^m + 1)}{N}}.$$

Now, for two hypothesis sets $H$ and $H'$ of functions mapping from $X$ to $\mathbb{R}$,

$$\mathcal{R}_N(\{\max(h, h') \mid h \in H, h' \in H'\}) \leq \mathcal{R}_N(H) + \mathcal{R}_N(H'), \tag{1}$$

where $\max(h, h')$ denotes the function $x \mapsto \max(h(x), h'(x))$ [Mohri et al., 2012]. Therefore,

$$\mathcal{R}_N(\Phi_j) \leq (n+1)^m \mathcal{R}_N(\mathcal{H}_j^1) \leq (n+1)^m \hat{H}_v(nH_{\overline{w}} + H_\lambda)\sqrt{\frac{2\log(n2^m + 1)}{N}},$$

which means that,

$$\mathcal{R}_N(\mathcal{L}_j) = \frac{1}{H_{\underline{w}}} \mathcal{R}_N(\Phi_j) \leq \frac{(n+1)^m \hat{H}_v(nH_{\overline{w}} + H_\lambda)}{H_{\underline{w}}} \sqrt{\frac{2\log(n2^m + 1)}{N}}$$

$$= O\left(\frac{n^m \hat{H}_v(nH_{\overline{w}} + H_\lambda)}{H_{\underline{w}}} \sqrt{\frac{m \log n}{N}}\right).$$

$\square$

**Lemma 3.**

$$\mathcal{R}_N(\mathcal{L}_{n+1}) = O\left(\frac{n^{m+2}(H_{\overline{w}}H_v + H_\lambda)}{H_{\underline{w}}} \sqrt{\frac{m \log n}{N}} \left(\frac{n\hat{H}_v(nH_{\overline{w}} + H_\lambda)}{H_{\underline{w}}} + \sqrt{n^m \log N}\right)\right).$$

*Proof.* Recall that

$$rev_{A,n+1}(\vec{v}) = -\sum_{i=1}^{(n+1)^m} \left(\sum_{j=1}^{n} \frac{1}{w_j} \sum_{\ell \neq j} w_\ell v_\ell\left(o_{i,\ell}\right) + \lambda_i\right) \mathbf{1}_{\vec{o}_i = \vec{o}_A^*(\vec{v})}$$

and $\mathcal{L}_{n+1} = \left\{ rev_{A,j} \mid (w_1, \ldots, w_n, \lambda_1, \ldots, \lambda_{(n+1)^m}), H_{\underline{w}} \leq |w_i| \leq H_{\overline{w}}, |\lambda_i| \leq H_\lambda \right\}$. In essence, for a fixed AMA $A$, $rev_{A,n+1}$ is the sum of $(n+1)^m$ linear functions multiplied by a binary function. In this proof, we will analyze the complexity of the linear and binary functions separately. Then, we can combine our analyses by making use of the following helpful lemma, which is similar to Lemma 3 in DeSalvo et al. [2015].

**Lemma 4.** *Let $\mathcal{F}$ be a family of functions mapping $\mathcal{X}$ to $[-c, c]$, let $\mathcal{G}$ be a a family of functions mapping $\mathcal{X}$ to $\{0, 1\}$, and let $\mathcal{H} = \{fg \mid f \in \mathcal{F}, g \in \mathcal{G}\}$. Then*

$$\mathcal{R}_N(\mathcal{H}) \leq (c+1)(\mathcal{R}_N(\mathcal{F}) + \mathcal{R}_N(\mathcal{G})).$$

*Proof of Lemma 4.* Notice that for any $f \in \mathcal{F}, g \in \mathcal{G}$, we have that $fg = \frac{1}{4}[(f+g)^2 - (f-g)^2]$. For $x \in [-c, c+1]$, the function $x \mapsto \frac{1}{4}x^2$ is $\frac{1}{2}(c+1)$-Lipschitz. The same holds for $x \in [-c-1, c]$. Therefore, by Talagrand's lemma (see Mohri et al. [2012], for example), we have that

$$\widehat{\mathcal{R}}_S(\mathcal{H}) \leq \frac{1}{2}(c+1)[\widehat{\mathcal{R}}_S(\mathcal{F} + \mathcal{G}) + \widehat{\mathcal{R}}_S(\mathcal{F} - \mathcal{G})] \leq (c+1)(\widehat{\mathcal{R}}_S(\mathcal{F}) + \widehat{\mathcal{R}}_S(\mathcal{G})).$$

Therefore, $\mathcal{R}_N(\mathcal{H}) \leq (c+1)(\mathcal{R}_N(\mathcal{F}) + \mathcal{R}_N(\mathcal{G}))$. $\square$

To use Lemma 4, we first define a set of functions for each $i \in [(n+1)^m]$ which will function as the set of real-valued functions in the lemma statement. In particular, let

$$\mathcal{F}_i = \left\{ f_{A,i} \mid A = (w_1, \ldots, w_n, \lambda_1, \ldots, \lambda_{(n+1)^m}), H_{\underline{w}} \leq |w_j| \leq H_{\overline{w}}, |\lambda_j| \leq H_\lambda \right\},$$

where

$$f_{A,i}(\vec{v}) = \sum_{j=1}^{n} \frac{1}{w_j} \sum_{\ell \neq j} w_\ell v_\ell(o_{i,\ell}) + \lambda_i.$$

As in the proof of Lemma 2, we can write each $f_{A,i}$ as a linear function from $\mathbb{R}^{n2^m+1}$ to $\mathbb{R}$ as follows. Let $h_{A,i}(\vec{v}, 1) = (\vec{v}, 1) \cdot \vec{a}_{A,i}$, where

$$\vec{a}_{A,i}[\ell] = \begin{cases} w_t \sum_{s \neq t} \frac{1}{w_s} & \text{if } \ell = 2^m(t-1) + \sigma(i,t) \\ \lambda_i \sum_{i=1}^{n} \frac{1}{w_i} & \text{if } \ell = n2^m + 1 \\ 0 & \text{otherwise} \end{cases}.$$

Then $f_{A,i}(\vec{v}) = h_{A,i}(\vec{v}, 1)$. As before, we have that $||(\vec{v}, 1)||_\infty \leq \max\{H_v, 1\} = \hat{H}_v$. Moreover, $||\vec{a}_{A,i}||_1 \leq \frac{n}{H_{\underline{w}}}(nH_{\overline{w}} + H_\lambda)$. Again, using the same $\ell_1$-norm Rademacher complexity bound for linear functions as in Lemma 2, we have that

$$\mathcal{R}_N(\mathcal{F}_i) \leq \frac{n\hat{H}_v}{H_{\underline{w}}}(nH_{\overline{w}} + H_\lambda)\sqrt{\frac{2\log(n2^m + 1)}{N}}. \tag{2}$$

Now, we define a set of functions $\mathcal{G}_i$ for each $i \in [(n+1)^m]$ which will function as the set of binary-valued functions in the statement of Lemma 4. Let

$$\mathcal{G}_i = \left\{ g_{A,i} \mid A = (w_1, \ldots, w_n, \lambda_1, \ldots, \lambda_{(n+1)^m}), H_{\underline{w}} \leq |w_j| \leq H_{\overline{w}}, |\lambda_j| \leq H_\lambda \right\},$$

where $g_{A,i}(\vec{v}) = 1$ if and only if $i = \vec{o}_A^*(\vec{v})$, i.e.

$$g_{A,i}(\vec{v}) = \begin{cases} 1 & \text{if } i = \underset{i \in [(n+1)^m]}{\text{argmax}} \left\{ \sum_{j=1}^{n} w_j v_j(o_{i,j}) + \lambda_i \right\} \\ 0 & \text{otherwise} \end{cases}.$$

If we multiply $f_{A,i}(\vec{v})$ and $g_{A,i}(\vec{v})$, then we get $\left( \sum_{j=1}^{n} \frac{1}{w_j} \sum_{\ell \neq j} w_\ell v_\ell(o_{i,\ell}) + \lambda_i \right) \mathbf{1}_{\vec{o}_i = \vec{o}_A^*(\vec{v})}$, which is one of the summands in $rev_{A,n+1}(\vec{v})$.

We now bound the Rademacher complexity of the class $\mathcal{G}_i$ so that we can apply Lemma 4. Notice that we can also write each function $g_{A,i}(\vec{v})$ as an intersection of $(n+1)^m - 1$ binary-valued functions $\{c_\ell\}$, where $\ell \in [(n+1)^m] \setminus \{i\}$, as follows.

$$c_\ell(\vec{v}) = \begin{cases} 1 & \text{if } \sum_{j=1}^{n} w_j v_j(o_{i,j}) + \lambda_i \geq \sum_{j=1}^{n} w_j v_j(o_{\ell,j}) + \lambda_\ell \\ 0 & \text{otherwise} \end{cases}. \tag{3}$$

Indeed, $c_\ell(\vec{v}) = 1$ for all $\ell \neq i$ if and only if $g_{A,i}(\vec{v}) = 1$, i.e.

$$i = \underset{i \in [(n+1)^m]}{\text{argmax}} \left\{ \sum_{j=1}^{n} w_j v_j(o_{i,j}) + \lambda_i \right\}.$$

Each function $c_\ell$ can be written as a linear separator over $\mathbb{R}^{n2^m}$, so the VC dimension of $\{c_\ell\}$ is $n2^m + 1$. This allows us to use Lemma 3.2.3 from Blumer et al. [1989] to bound the VC dimension of $\mathcal{G}_i$.

**Lemma 5** (Lemma 3.2.3 from Blumer et al. [1989]). *Let $C$ be a concept class of finite VC dimension $d \geq 1$. For all $s \geq 1$, let $C_s\{\cap_{i=1}^{s} c_i \mid c_i \in C, 1 \leq i \leq s\}$. Then for all $s \geq 1$, the VC dimension of $C_s$ is less than $2ds\log(3s)$.*

Therefore, the VC dimension of $\mathcal{G}_i$ is less than $2(n2^m + 1)(n + 1)^m \log(3 \cdot (n + 1)^m) = O(mn^m \log n)$. By Corollary 3.1 in Mohri et al. [2012], we have that

$$\mathcal{R}_N(\mathcal{G}_i) = O\left(\sqrt{\frac{mn^m \log n \log N}{N}}\right). \tag{4}$$

Putting Equations (2) and (4) together with Lemma 4, we conclude that if $\mathcal{H}_i = \{f_{A,i} g_{A,i} \mid f_{A,i} \in \mathcal{F}_i, g_{A,i} \in \mathcal{G}_i\}$, then

$$\mathcal{R}_N(\mathcal{H}_i) = O\left(\left(\frac{n^2(H_{\overline{w}} H_v + H_\lambda)}{H_{\underline{w}}}\right)\left(\frac{n\hat{H}_v}{H_{\underline{w}}}(nH_{\overline{w}} + H_\lambda)\sqrt{\frac{m \log n}{N}} + \sqrt{\frac{mn^m \log n \log N}{N}}\right)\right).$$

This follows from Lemma 4, since the range of any function in $\mathcal{F}_i$ is $[0, n(n-1)(H_{\overline{w}} H_v + H_\lambda)/H_{\underline{w}}]$.

Finally, since

$$rev_{A,n+1}(\vec{v}) = -\sum_{i=1}^{(n+1)^m} f_{A,i}(\vec{v}) g_{A,i}(\vec{v}),$$

we have that

$$\begin{aligned}
&\mathcal{R}_N(\mathcal{L}_{n+1}) \\
&= O\left(n^m \left(\frac{n^2(H_{\overline{w}} H_v + H_\lambda)}{H_{\underline{w}}}\right)\left(\frac{n\hat{H}_v}{H_{\underline{w}}}(nH_{\overline{w}} + H_\lambda)\sqrt{\frac{m \log n}{N}} + \sqrt{\frac{mn^m \log n \log N}{N}}\right)\right).
\end{aligned}$$

By rearranging terms, we get the desired result. $\qquad\square$

## B.2 Lower Bound on Sample Complexity for $\lambda$-Auctions

In this section, we show that it is not possible to learn over the set of $\lambda$-auction revenue functions under an arbitrary distribution with subexponential sample complexity. Since $\lambda$-auctions are a subset of AMAs, this lower bound applies to AMAs as well. In particular, we prove Theorem 8, which states that no algorithm can learn over the class of $n$-bidder, $m$-item $\lambda$-auction revenue functions with sample complexity $o(n^m)$. This holds even when the bidders' valuation functions are additive.

To prove Theorem 8, we construct a set $V$ of $n$-bidder, $m$-item valuation functions taking values in $\{0, 1\}$ where, under each valuation function, each bidder is interested in a specific subset of items, and these subsets are all pairwise disjoint. Moreover, $|V| = n^m - n$. The high level idea is to show that for any subset $H$ of $V$, there exists a $\lambda$-auction that has high revenue over valuation functions in $H$, but low revenue on the valuation functions in $V \setminus H$. Theorem 7 describes $V$ in more detail. Now suppose that the distribution over the bidders' valuation functions is the uniform distribution over $V$. This means that if a learning algorithm's input samples consist of only a small subset of $V$, then we cannot guarantee that any output revenue function will achieve average revenue over the sample which is close to its expected revenue over the distribution, as we require. This immediately implies hardness for learning over the uniform distribution on $V$. See Theorem 8 for the formal proof.

We now present Theorem 7, wherein we describe the set $V$ of valuation functions which we will use to prove Theorem 8.

**Theorem 7.** *For any $n, m \geq 2$ and any $\gamma \in (0, 1)$, there exists a set of $N = n^m - n$ $n$-bidder, $m$-item additive valuation functions $V = \{\vec{v}^1, \dots, \vec{v}^N\}$ such that for any $H \subseteq V$, there exists a $\lambda$-auction $A_H$ with revenue 0 on $\vec{v}^i$ if $\vec{v}^i \notin H$ and revenue at least $2 - 2\gamma$ on $\vec{v}^i$ otherwise.*

*Proof.* We define the set $V = \{\vec{v}^1, \dots, \vec{v}^N\}$ of $n$-bidder, $m$-item additive valuation functions, where $\vec{v}^j = \left(v_1^j(\{1\}), \dots, v_1^j(\{m\}), \dots, v_n^j(\{1\}) \dots, v_n^j(\{m\})\right)$, with $N = n^m - n$. Recall that every allocation vector $\vec{o}_j$ is written as $(o_{j,1}, \dots, o_{j,n})$ where $o_{j,1}, \dots, o_{j,n}$ are disjoint subsets of the $m$ items being auctioned. First, let $\hat{o}_j$ be the allocation where Bidder $j$ receives all $m$ items. Next, let $\tilde{o}_1, \dots, \tilde{o}_N$ be a fixed ordering of the $n^m - n$ allocations where all $m$ goods are allocated except $\{\hat{o}_1, \dots, \hat{o}_n\}$. Let the bundles allocated to the $n$ bidders in $\tilde{o}_\ell$ be $(\tilde{o}_{\ell,1}, \dots, \tilde{o}_{\ell,n})$ and let $N_\ell$ be the

set of bidders who are allocated some item in allocation $\tilde{o}_\ell$. In other words, $N_\ell = \{j \mid \tilde{o}_{\ell,j} \neq \emptyset\}$. For a sanity check, notice that $\bigcup_{i=1}^n \tilde{o}_{\ell,i}$ is the grand bundle.

We will now define the valuation vectors $\{\vec{v}^1, \ldots, \vec{v}^N\}$ in terms of this set of special allocations $\{\tilde{o}_1, \ldots, \tilde{o}_N\}$. Specifically, we define $\vec{v}^\ell$ for $\ell \in [N]$ as follows.

If $i \notin N_\ell$ (i.e. $\tilde{o}_{\ell,j} = \emptyset$), set $v_i^\ell(\{j\}) = 0$ for all $j \in [m]$. Otherwise, set

$$v_i^\ell(\{j\}) = \begin{cases} 0 & \text{if } j \notin \tilde{o}_{\ell,i} \\ 1 & \text{if } j \in \tilde{o}_{\ell,i} \end{cases}.$$

We proceed to prove that for any subset $H \subseteq V$, there exists a $\lambda$-auction with 0 revenue on all valuation functions in $V \setminus H$ and at least $2 - 2\gamma$ revenue on all valuation functions in $H$. To define this $\lambda$-auction, we set the $\lambda$ terms such that

$$\lambda(\vec{o_j}) = \begin{cases} 0 & \text{if } \vec{o}_j = \tilde{o}_\ell \text{ for some } \vec{v}^\ell \in H \\ 1 - \gamma & \text{otherwise} \end{cases}.$$

**Lemma 6.** *If $\vec{v}^\ell \in H$, then the revenue on $\vec{v}^\ell$ is at least $2 - 2\gamma$.*

*Proof of Lemma 6.* First, note that $\sum_{i=1}^n v_i^\ell(\tilde{o}_{\ell,i}) + \lambda(\tilde{o}_\ell) = m$, and for all allocations $\vec{o}_j \neq \tilde{o}_\ell$, $\sum_{i=1}^n v_i^\ell(o_{j,i}) + \lambda(\vec{o}_j) \leq m - 1 + 1 - \gamma < m$. Therefore, the $\lambda$-auction allocation is $\tilde{o}_\ell$.

In order to analyze the revenue of this $\lambda$-auction, we must understand the payments of each bidder, which means that we must investigate what the outcome of this $\lambda$-auction would be without any one bidder's participation. To this end, suppose $i \in N_\ell$, so Bidder $i$ is allocated some item in $\tilde{o}_\ell$, i.e. $\tilde{o}_{\ell,i} \neq \emptyset$. Then $\sum_{j \neq i} v_j^\ell(\tilde{o}_{\ell,j}) + \lambda(\tilde{o}_\ell) = m - |\tilde{o}_{\ell,i}|$ because Bidder $i$'s valuation for the bundle $\tilde{o}_{\ell,i}$ is exactly $|\tilde{o}_{\ell,i}|$.

By construction, no bidder receives all $m$ items in $\tilde{o}_\ell$, so we know that there exists some $i' \in N_\ell, i' \neq i$. With this fact in mind, let $\vec{o}_{-i}$ be the allocation where all bidders in $N_\ell$ are allocated the same items as they are in $\tilde{o}_\ell$ and Bidder $i$ receives the empty set. This is one possible allocation of the $\lambda$-auction without Bidder $i$'s participation, and therefore the social welfare of the other bidders will be at least as high under this allocation as it would be in the true allocation of the $\lambda$-auction without Bidder $i$'s participation. By construction, $\lambda(\vec{o}_{-i}) = 1 - \gamma$. Therefore, $\sum_{\ell \neq i} v_j^\ell(o_{-i,j}) + \lambda(\vec{o}_{-i}) = m - |\tilde{o}_{\ell,i}| + 1 - \gamma$ which means that Bidder $i$ must pay at least $(m - |\tilde{o}_{\ell,i}| + 1 - \gamma) - (m - |\tilde{o}_{\ell,i}|) = 1 - \gamma$. We know that $|N_\ell| \geq 2$, i.e. there are at least 2 bidders who receive a non-empty bundle and therefore must pay at least $1 - \gamma$, so the revenue of this $\lambda$-auction is at least $2 - 2\gamma$. $\qquad\square$

**Lemma 7.** *If $\vec{v}^\ell \notin H$, then the revenue on $\vec{v}^\ell$ is 0.*

*Proof of Lemma 7.* First, note that $\sum_{i=1}^n v_i^\ell(\tilde{o}_{\ell,i}) + \lambda(\tilde{o}_\ell) = m + 1 - \gamma$, and for all allocations $\vec{o}_j \neq \tilde{o}_\ell$, $\sum_{i=1}^n v_i^\ell(\vec{o}_{j,i}) + \lambda(\vec{o}_j) \leq m - 1 + 1 - \gamma < m$, so the $\lambda$-auction allocation is $\tilde{o}_\ell$. Now, suppose $i \in N_\ell$. Then $\sum_{j \neq i} v_j^\ell(\tilde{o}_{\ell,j}) + \lambda(\tilde{o}_\ell) = m - |\tilde{o}_{\ell,i}| + 1 - \gamma$. Since Bidder $i$ is the only bidder with nonzero valuations for the items in $\tilde{o}_{\ell,i}$ under $\vec{v}^\ell$, any allocation $\vec{o}_{-i}$ without his participation will have social welfare at most $\sum_{j \neq i} v_j^\ell(o_{-i,j}) + \lambda(\vec{o}_{-i}) \leq m - |\tilde{o}_{\ell,i}| + 1 - \gamma$. Therefore, Bidder $i$ pays nothing.

Of course, for any Bidder $i \notin N_\ell$, her presence in the auction makes no difference on the resulting allocation because her valuation function under $\vec{v}^\ell$ is 0 on all items, so she pays nothing as well. Therefore, the revenue on $\vec{v}^\ell$ is 0. $\qquad\square$

Putting Lemmas 6 and 7 together, we have the desired result. $\qquad\square$

We now use Theorem 7 to prove Theorem 8.

**Theorem 8.** *Let $\mathcal{ALG}$ be an arbitrary learning algorithm that uses only a polynomial number of training samples drawn i.i.d. from the underlying distribution and produces a $\lambda$-auction. For any $\epsilon \in (0, 1)$, there exists a distribution $\mathcal{D}$ and a $\lambda$-auction $A^*$ such that, with probability 1 (over the draw of the set of training samples $\mathcal{S}$),*

$$\frac{1}{|\mathcal{S}|} \sum_{\vec{v} \in \mathcal{S}} rev_{A^*}(\vec{v}) - \mathbb{E}_{\vec{v} \sim \mathcal{D}}[rev_{A^*}(\vec{v})] > \epsilon.$$

*Proof.* Let $\gamma = 1 - \epsilon$ and let $V$ be the set of valuation functions proven to exist in Theorem 7 corresponding to $\gamma$ (i.e. for any $H \subseteq V$, there exists a $\lambda$-auction $A_H$ with revenue 0 on $\vec{v}$ if $\vec{v} \in H$ and revenue at least $2 - 2\gamma$ on $\vec{v}$ otherwise). Let $\mathcal{D}$ be the uniform distribution on $V$.

Suppose that $\mathcal{ALG}$ uses a set $\mathcal{S}$ of $\ell \leq n^c$ samples, where $c$ is a constant. Of course, $\mathcal{S} \subseteq V$, so let $A^*$ be the $\lambda$-auction with 0 revenue on every valuation function not in the sample and revenue at least $2 - 2\gamma$ on every valuation function in the sample. We know that $A^*$ exists due to Theorem 7.

Notice that the average empirical revenue of $A^*$ on $\mathcal{S}$ is at least $2 - 2\gamma$. Meanwhile, the probability, on a random draw $\vec{v} \sim \mathcal{D}$ that $rev_{A^*}(\vec{v})$ is 0 is exactly the probability that $\vec{v} \notin \mathcal{S}$. Given that the set of training examples has measure $\frac{n^c}{n^m - n} < \frac{1}{2}$, we have that

$$\frac{1}{|\mathcal{S}|} \sum_{\vec{v} \in \mathcal{S}} rev_{A^*}(\vec{v}) - \mathbb{E}_{\vec{v} \sim \mathcal{D}}[rev_{A^*}(\vec{v})] \geq 2 - 2\gamma - (2 - 2\gamma) \mathbb{P}_{\vec{v} \sim \mathcal{D}}[\vec{v} \in \mathcal{S}]$$
$$> 2 - 2\gamma - (1 - \gamma)$$
$$= 1 - \gamma$$
$$= \epsilon,$$

as desired. □

## B.3  Lower Bound on Sample Complexity for VVCAs

In this section, we prove that it is not possible to learn over the set of VVCA revenue function under and arbitrary distribution with subexponential sample complexity. In particular, we prove that no algorithm can learn over the class of $n$-bidder, $m$-item VVCA revenue functions with sample complexity $o(2^m)$. This holds even when the bidders' valuation functions are additive.

The format of this proof similar to that of Theorem 8. Namely, we construct a set $V$ of $n$-bidder, $m$-item valuation functions such that $|V| = 2^m - 2$. We then show that for any subset $H$ of $V$, there exists a VVCA that has high revenue over valuation functions in $H$, but low revenue on the valuation functions in $V \setminus H$. The set $V$ is described in more detail in Theorem 9. As described in Theorem 8, this immediately implies hardness for learning over the uniform distribution on $V$. Given the parallel proof structure, we present Theorem 9 and refer the reader to Theorem 8 to see how it implies hardness for learning.

**Theorem 9.** *For any $m \geq 2$ and any $\gamma \in (0, 1)$, there exists a set of $N = 2^m - 2$ 2-bidder additive valuation functions $V = \{\vec{v}^1, \ldots, \vec{v}^N\}$ such that for any $H \subseteq V$, there exists a VVCA with revenue 0 on $\vec{v}^i$ if $\vec{v}^i \in V$ and revenue $1 - \gamma$ on $\vec{v}^i$ if $\vec{v}^i \notin V$.*

*Proof.* We define the set $V = \{\vec{v}^1, \ldots, \vec{v}^N\}$ of 2-bidder valuation functions, where $\vec{v}^j = (v_1^j(\{1\}), \ldots, v_1^j(\{m\}), v_2^j(\{1\}) \ldots, v_2^j(\{m\}))$, with $N = 2^m - 2$. Recall that every allocation vector $\vec{o}_j$ can be written as $(o_{j,1}, o_{j,2})$ where $o_{j,1}$ and $o_{j,2}$ are disjoint subsets of the $m$ items being auctioned. In order to define the valuation functions in $V$, we define $\tilde{b}_1, \ldots, \tilde{b}_N$ to be a arbitrary, fixed ordering of all subsets of $[m]$ except the empty set and the grand bundle. In other words, $\tilde{b}_1, \ldots, \tilde{b}_N$ is an ordering of $2^{[m]} \setminus \{\emptyset, [m]\}$. We will define each valuation function in $V$ in terms of this ordering. In particular, let $\tilde{o}_\ell = (\tilde{b}_\ell^c, \tilde{b}_\ell)$ be the allocation where Bidder 1 receives $\tilde{b}_\ell^c$ and Bidder 2 receives $\tilde{b}_\ell$. Finally, let $\vec{v}^\ell$ for $\ell \in [N]$ be defined as follows.

$$v_1^\ell(\{i\}) = \begin{cases} 1 & \text{if } i \in \tilde{b}_\ell^c \\ 0 & \text{otherwise} \end{cases}$$

and

$$v_2^\ell(\{i\}) = \begin{cases} 1 & \text{if } i \in \tilde{b}_\ell \\ 0 & \text{otherwise} \end{cases}.$$

Clearly, if $w_1 = w_2 = 1$ and $\lambda_1(\vec{o}) = \lambda_2(\vec{o}) = 0$ for all $\vec{o} \in \mathcal{O}$, then the VVCA allocation on any $\vec{v}^\ell \in S$ is the one in which Bidder 2 receives $\tilde{b}_\ell^c$ and Bidder 1 receives $\tilde{b}_\ell$. This has a social welfare of $m$, whereas any other allocation has a social welfare at most $m - 1$.

We claim that for any $H \subseteq V$, there exists a VVCA with revenue 0 on $\vec{v}^i$ if $\vec{v}^i \in H$ and revenue $1 - \gamma$ on $\vec{v}^i$ if $\vec{v}^i \notin H$. The VVCA has bidder weights $w_1 = w_2 = 1$, and for all $\vec{v}^\ell \in H$, we set $\lambda_1(\tilde{o}_\ell) = c_{1,\tilde{b}_\ell^c} = c_{2,\tilde{b}_\ell} = \lambda_2(\tilde{o}_\ell) = 0$. Otherwise, we set $\lambda_i(\vec{o}) = (1-\gamma)/2$ for each $i \in \{1, 2\}$.

**Lemma 8.** *If $\vec{v}^\ell \in H$, then the revenue on $\vec{v}^\ell$ is $1 - \gamma$.*

*Proof of Lemma 8.* First, note that $v_1(\tilde{b}_\ell^c) + v_2(\tilde{b}_\ell) + \lambda_1(\tilde{o}_\ell) + \lambda_2(\tilde{o}_\ell) = m$, and for all allocations $\vec{o}_j \neq \tilde{o}_\ell$, $v_1(o_{j,1}) + v_2(o_{j,2}) + \lambda_1(\vec{o}_j) + \lambda_2(\vec{o}_j) \leq m - 1 + 1 - \gamma$. Therefore, the VVCA allocation is $\tilde{o}_\ell$. However, this is neither Bidder 1 nor Bidder 2's favorite weighted allocation, since $v_1(\tilde{b}_\ell^c) + \lambda_1(\tilde{o}_\ell) = |\tilde{b}_\ell^c| < v_1([m]) + c_{1,[m]} = |\tilde{b}_\ell^c| + (1-\gamma)/2$ and $v_2(\tilde{b}_\ell) + \lambda_2(\tilde{o}_\ell) = |\tilde{b}_\ell| < v_2([m]) + c_{2,[m]} = |\tilde{b}_\ell| + (1-\gamma)/2$. This follows from the fact that $\tilde{b}_\ell \neq [m]$ and $\tilde{b}_\ell^c \neq [m]$ for all $\ell \in [N]$, it must be that $\lambda_1([m]) = \lambda_2([m]) = (1-\gamma)/2$.

Since $|\tilde{b}_\ell^c|$ and $|\tilde{b}_\ell|$ are Bidder 1 and 2's highest valuations for any allocation, respectively, and because $(1-\gamma)/2$ is the highest value of any $\lambda$ term, $v_1([m]) + c_{1,[m]}$ and $v_2([m]) + c_{2,[m]}$ are the maximum weighted valuation that either bidder has for any allocation under this VVCA. Therefore, the revenue of this VVCA on $\vec{v}_\ell$ is $|\tilde{b}_\ell| + |\tilde{b}_\ell^c| + 1 - \gamma - |\tilde{b}_\ell| - |\tilde{b}_\ell^c| = 1 - \gamma$. $\qquad\square$

**Lemma 9.** *If $\vec{v}^\ell \notin H$, then the revenue on that valuation function pair is 0.*

*Proof of Lemma 9.* First, note that $v_1(\tilde{b}_\ell^c) + v_2(\tilde{b}_\ell) + \lambda_1(\tilde{o}_\ell) + \lambda_2(\tilde{o}_\ell) = m + 1 - \gamma$, and for all allocations $\vec{o}_j \neq \tilde{o}_\ell$, $v_1(o_{j,1}) + v_2(o_{j,2}) + \lambda_1(\vec{o}_j) + \lambda_2(\vec{o}_j) \leq m - 1 + 1 - \gamma < m + 1 - \gamma$, so the AMA allocation is $\tilde{o}_\ell$. Moreover, $v_1(\tilde{b}_\ell^c) + \lambda_1(\tilde{o}_\ell) = |\tilde{b}_\ell^c| + (1-\gamma)/2 \geq v_1(o_{j,1}) + \lambda_1(\vec{o}_j)$ and $v_2(\tilde{b}_\ell) + \lambda_2(\tilde{o}_\ell) = |\tilde{b}_\ell| + (1-\gamma)/2 \geq v_2(o_{j,2}) + \lambda_2(\vec{o}_j)$ for all allocations $\vec{o}_j \in \mathcal{O}$. Therefore, both bidders receive one of their favorite weighted allocations, so the revenue is 0. $\qquad\square$

$\qquad\qquad\qquad\qquad\qquad\qquad\qquad\qquad\qquad\qquad\qquad\qquad\qquad\qquad\qquad\qquad\square$

## C  Proofs from Section 3.2

*Proof of Lemma 1.* We will show that $rev_{\vec{v}}$ can be decomposed into simple components, each of which can be easily analyzed on its own, and by combining these analyses, we prove the lemma statement. To this end, recall that under the VCG mechanism, each winning bidder pays her bid minus a "rebate" equal to the increase in welfare attributable to her presence in the auction. In a $c$-MBA, each winning bidder pays the boosted version of this amount. In other words, suppose $\vec{o}^*$ is the resulting allocation of a certain $c$-MBA $A$ and $\vec{o}_{-i}$ is the boosted social-welfare maximizing allocation without Bidder $i$'s participation. More explicitly, $\vec{o}^* = \max_{\vec{o}_j} \left\{ \sum_{i=1}^n v_i(o_{j,i}) + \lambda(\vec{o}_j) \right\}$ and $\vec{o}_{-i} = \max_{\vec{o}_j} \left\{ \sum_{k \neq i} v_k(o_{j,k}) + \lambda(\vec{o}_j) \right\}$, where $\lambda(\vec{o}_j)$ is set according to the MBA allocation boosting rule for all $\vec{o}_j$. Then Bidder $i$ pays

$$p_{i,\vec{v}}(c) = v_i(o_i^*) - \left[ \sum_{j=1}^n v_j(o_j^*) + \lambda(\vec{o}^*) - \left( \sum_{j \neq i} v_j(o_{-i,j}) + \lambda(\vec{o}_{-i}) \right) \right],$$

where $c$ is the parameter of the $c$-MBA, factored into the $\lambda$ terms. This means that

$$rev_{\vec{v}}(c) = \sum_{i=1}^n p_{i,\vec{v}}(c) = (1-n) \sum_{i=1}^n v_i(o_i^*) - n\lambda(\vec{o}^*) + \sum_{i=1}^n \sum_{j \neq i} v_j(o_{-i,j}) + \lambda(\vec{o}_{-i}).$$

The revenue function can be split into $n + 1$ functions:

$$f_{i,\vec{v}}(c) = \sum_{j \neq i} v_j(o_{-i,j}) + \lambda(\vec{o}_{-i}) \text{ for } i \in \{1, \dots, n\}$$

and

$$g_{\vec{v}}(c) = (1-n) \sum_{i=1}^n v_i(o_i^*) - n\lambda(\vec{v}^*).$$

We claim that $f_{i,\vec{v}}(c)$ is continuous for all $i$, whereas $g_{\vec{v}}(c)$ has at most one discontinuity. This means that $rev_{\vec{v}}(c) = \sum_{i=1}^{n} f_{i,\vec{v}}(c) + g_{\vec{v}}(c)$ has at most one discontinuity as well. Moreover, the slope of $\sum_{i=1}^{n} f_{i,\vec{v}}(c)$ is between zero and $n$, whereas the slope of $g_{\vec{v}}(c)$ is zero until its discontinuity, and then is $-n$. Therefore, the slope of $rev_{\vec{v}}(c)$ is at least zero before its discontinuity and at most zero after its discontinuity. This is enough to prove the lemma statement.

To see why these properties are true for the functions $f_{i,\vec{v}}(c)$, first let $\vec{o}^{1}_{-i}$ be the VCG allocation without Bidder $i$'s participation. In other words, $\vec{o}^{1}_{-i} = \max_{\vec{o}_j} \left\{ \sum_{k \neq i} v_k(o_{j,k}) \right\}$. If one bidder is allocated the grand bundle in outcome $\vec{o}^{1}_{-i}$, then this allocation will only be more valuable as $c$ grows, so $\vec{o}^{1}_{-i} = \max_{\vec{o}_j} \left\{ \sum_{k \neq i} v_k(o_{j,k}) + \lambda(\vec{o}_j) \right\}$ for all values of $c$, which means that $f_{i,\vec{v}}(c) = \sum_{j \neq i} v_j\left(o^{1}_{-i,j}\right) + \lambda\left(\vec{o}^{1}_{-i}\right) = \sum_{j \neq i} v_j\left(o^{1}_{-i,j}\right) + c$ for all values of $c$ as well. Clearly, in this case, $f_{i,\vec{v}}(c)$ is increasing and continuous. Otherwise, there exists some value $c_i$ such that

$$\sum_{j \neq i} v_j\left(o^{1}_{-i,j}\right) + \lambda\left(\vec{o}^{1}_{-i}\right) = \sum_{j \neq i} v_j\left(o^{1}_{-i,j}\right) \geq \max_{k \neq i}\{v_k([m])\} + c \qquad \text{if } c \leq c_i$$

$$\sum_{j \neq i} v_j\left(o^{1}_{-i,j}\right) < \max_{k \neq i}\{v_k([m])\} + c \qquad \text{if } c > c_i.$$

This means that $\vec{o}^{1}_{-i}$ is the allocation of the $c$-MBA without Bidder $i$'s participation for $c \leq c_i$, and the allocation of the $c$-MBA without Bidder $i$'s participation for $c > c_i$ is the one where the highest bidder for the grand bundle (excluding Bidder $i$) wins the grand bundle. Therefore,

$$f_{i,\vec{v}}(c) = \begin{cases} \sum_{j \neq i} v_j\left(o^{1}_{-i,j}\right) & \text{if } c \leq c_i \\ \max_{k \neq i}\{v_k([m])\} + c & \text{if } c > c_i. \end{cases}$$

Notice that $\sum_{j \neq i} v_j\left(o^{1}_{-i,j}\right) = \max_{k \neq i}\{v_k([m])\} + c_i$, so $f_{i,\vec{v}}(c)$ is continuous. Finally, it is clear that the slope of each $f_{i,\vec{v}}(c)$ is between 0 and 1, so the slope of $\sum_{i=1}^{n} f_{i,\vec{v}}(c)$ is between 0 and $n$.

Similarly, let $\vec{o}^{1}$ be the allocation of the VCG mechanism run on $\vec{v}$. Then there exists some $c^*$ such that $\vec{o}^{1}$ is the allocation of the $c$-MBA for $c \leq c^*$ and the allocation of the $c$-MBA for $c > c_i$ is the one where the highest bidder for the grand bundle wins the grand bundle. More explicitly,

$$\sum_{i=1}^{n} v_i\left(o^{1}_i\right) + \lambda\left(\vec{o}^{1}\right) = \sum_{i=1}^{n} v_i\left(o^{1}_i\right) \geq \max\{v_k([m])\} + c \qquad \text{if } c \leq c^*$$

$$\sum_{i=1}^{n} v_i\left(o^{1}_i\right) < \max\{v_k([m])\} + c \qquad \text{if } c > c^*.$$

Therefore,

$$g_{\vec{v}}(c) = \begin{cases} (1-n)\sum_{i=1}^{n} v_i\left(o^{1}_i\right) & \text{if } c \leq c^* \\ (1-n)\max\{v_k([m])\} - nc & \text{if } c > c^*. \end{cases}$$

Therefore, $g_{\vec{v}}(c)$ has at most one discontinuity, which falls at $c^*$. Moreover, the slope of $g_{\vec{v}}(c)$ is 0 for $c < c^*$ and $-n$ for $c > c^*$. As described, these properties of $f_{i,\vec{v}}(c)$ and $g_{\vec{v}}(c)$ are enough to show that the lemma statement holds. $\qquad\square$

*Proof of Theorem 3.* First, we show that the pseudo-dimension of the class of $n$-bidder, $m$-item MBAs is at most 2. Let $\mathcal{S} = \left\{\vec{v}^1, \ldots, \vec{v}^N\right\}$ of size $N$ be a set of $n$-bidder valuation functions that can be shattered by a set $C$ of $2^N$ MBAs. This means that there exist $N$ witnesses $z^1, \ldots, z^N$ such that each MBA in $C$ induces a binary labeling of the samples $\vec{v}^j$ of $\mathcal{S}$ (whether the revenue of the MBA on $\vec{v}^j$ is at least $z^j$ or strictly less than $z^j$). Since $\mathcal{S}$ is shatterable, we can thus label $\mathcal{S}$ in every possible way using MBAs in $C$.

Now, fix one sample $\vec{v}^i \in \mathcal{S}$ and consider $rev_{\vec{v}^i}(c)$. From Lemma 1, we know that there exists $c_i^* \in [0, \infty)$, such that $rev_{\vec{v}^i}(c)$ is non-decreasing on the interval $[0, c_i^*]$ and non-increasing on the interval $(c_i^*, \infty)$. Therefore, there exist two thresholds $t_i^1 \in [0, c_i^*]$ and $t_i^2 \in (c_i^*, \infty) \cup \{\infty\}$ such that $rev_{\vec{v}^i}(c)$ is below its threshold for $c \in [0, t_i^1)$, above its threshold for $c \in (t_i^1, t_i^2)$, and below its threshold for $c \in (t_i^2, \infty)$. Now, merge these thresholds for all $N$ samples on the real line and

| $c$ **value** | **Revenue on $\vec{v}^1$** | **Revenue on $\vec{v}^2$** |
|---|---|---|
| 0 | $0 \leq z^1$ | $2 \leq z^2$ |
| 1.5 | $3 \leq z^1$ | $5 > z^2$ |
| 2.5 | $5 > z^1$ | $4 \leq z^2$ |
| 2 | $4 > z^1$ | $6 > z^2$ |

Table 1: Example of a shattered set of size 2

consider the interval $(t_1, t_2)$ between two adjacent thresholds. The binary labeling of the samples in $\mathcal{S}$ on this interval is fixed. In other words, for any sample $\vec{v}^j \in \mathcal{S}$, $rev_{\vec{v}^j}(c)$ is either at least $z^j$ or strictly less than $z^j$ for all $c \in (t_1, t_2)$. There are at most $2N + 1$ intervals between adjacent thresholds, so at most $2N + 1$ different binary labelings of $\mathcal{S}$. Since we assumed $\mathcal{S}$ is shatterable, it must be that $2^N \leq 2N + 1$, so $N \leq 2$.

Finally, we show that the pseudo-dimension of the class of $n$-bidder, $m$-item MBAs is at least 2 by constructing a set $\mathcal{S} = \{\vec{v}^1, \vec{v}^2\}$ that can be shattered by the set of MBAs. To construct this sample $\mathcal{S}$, let

$$v_1^1(b_i) = v_2^1(b_i) = \begin{cases} 0 & \text{if } |b_i| < \lfloor m/2 \rfloor \\ 3 & \text{if } \lfloor m/2 \rfloor \leq |b_i| \end{cases} \text{ and } v_1^2(b_i) = v_2^2(b_i) = \begin{cases} 0 & \text{if } |b_i| < \lfloor m/2 \rfloor \\ 3 & \text{if } \lfloor m/2 \rfloor \leq |b_i| < m \\ 4 & \text{if } |b_i| = m. \end{cases}$$

Finally, let Bidders 3 through $n$ have all-zero valuations in both $\vec{v}^1$ and $\vec{v}^2$.

Now, let $z^1 = 3$ and $z^2 = 4$. We define four MBAs parameterized by the coefficients $c_1 = 0, c_2 = 1.5, c_3 = 2.5, c_4 = 2$. It is easy to check that this set of MBAs shatters $\mathcal{S}$, witnessed by $z^1$ and $z^2$. For example, see Table 1. □

*Proof of Theorem 4.* This follows from standard pseudo-dimension sample complexity bounds (e.g. Theorem 19.2 of Anthony and Bartlett [2009]) and Theorem 3. □

*Proof of Theorem 5.* Let $\mathcal{S} = \{\vec{v}^1, \ldots, \vec{v}^N\}$ of size $N$ be a set of $n$-bidder valuation function samples that can be shattered by a set $C$ of $2^N$ MBARPs. This means that there exist $N$ witnesses $z^1, \ldots, z^N$ such that each MBARP in $C$ induces a binary labeling of the samples $\vec{v}^j$ in $\mathcal{S}$ (whether the revenue of the MBARP on $\vec{v}^j$ is greater than $z^j$ or at most $z^j$). Since $\mathcal{S}$ is shatterable, we can thus label $\mathcal{S}$ in every possible way using MBARPs in $C$.

This proof is similar to the proof of Theorem 3, where we split the real line into a set of intervals $\mathcal{I}$ such that for any $I \in \mathcal{I}$, the binary labeling of $\mathcal{S}$ by the $c$-MBA revenue function was fixed for all $c \in I$. In the case of MBARPs, however, the domain is $\mathbb{R}^{m+1}$, so we cannot split the domain into intervals in the same way. Instead, we show that we can split the domain into cells such that the binary labeling of $\mathcal{S}$ by the MBARP revenue function is fixed as we range over parameters in a single cell. In this way, we show that $N = O(m^3 \log n)$.

Now, fix $\vec{v}^t \in \mathcal{S}$. First, for each $T \subseteq [m]$, let $\mathcal{O}_T$ be the set of allocations where exactly the elements of $T$ are allocated, and let

$$\vec{o}^T = \underset{\vec{o} \in \mathcal{O}_T}{\arg\max} \left\{ \sum_{i=1}^{n} v_i^t(o_i) \right\}.$$

Notice that regardless of the reserve prices, if $T$ comprises of the items allocated in the allocation of an MBARP, then $\vec{o}^T$ will be the allocation. After all, if $(r_1, \ldots, r_m)$ are the reserve prices of an arbitrary MBARP, then it will always be the case that

$$\sum_{i=1}^{n} v_i^t(o_i^T) + \sum_{j \notin T} r_j \geq \sum_{i=1}^{n} v_i^t(o_i') + \sum_{j \notin T} r_j$$

for any allocation $\vec{o}' \in \mathcal{O}_T$ by definition of $\vec{o}^T$.

Now, consider an MBARP parameterized by $(c, r_1, \ldots, r_m)$. The allocation will be

$$\vec{o}^T = \operatorname{argmax} \left\{ \sum_{i=1}^{n} v_i^t \left( o_i^{[m]} \right) + c, \left\{ \sum_{i=1}^{n} v_i^t \left( o_i^T \right) + \sum_{j \notin T} r_j \right\}_{T \neq [m]} \right\}.$$

For any $T \subseteq [m]$, let $R_T^{\vec{v}^t}$ be the subset of $\mathbb{R}^{m+1}$ such that if an MBARP is parameterized by $(c, r_1, \ldots, r_m) \in R_T^{\vec{v}^t}$, then the allocation of the MBARP on $\vec{v}^t$ is $\vec{o}^T$. This means that if $T \neq [m]$

$$\sum_{i=1}^{n} v_i^t \left( o_i^T \right) + \sum_{j \notin T} r_j \geq \sum_{i=1}^{n} v_i^t \left( o_i^{T'} \right) + \sum_{j \notin T'} r_j \qquad \forall T' \notin \{T, [m]\} \text{ and}$$

$$\sum_{i=1}^{n} v_i^t \left( o_i^T \right) + \sum_{j \notin T} r_j \geq \sum_{i=1}^{n} v_i^t \left( o_i^{[m]} \right) + c.$$

In other words, $(c, r_1, \ldots, r_m) \in R_T^{\vec{v}^t}$ if and only if it falls in the intersection of $2^m - 1$ halfspaces:

$$\sum_{j \notin T} r_j - \sum_{j \notin T'} r_j \geq \sum_{i=1}^{n} v_i^t \left( o_i^{T'} \right) - v_i^t \left( o_i^T \right) \qquad \forall T' \notin \{T, [m]\}$$

$$\sum_{j \notin T} r_j - c \geq \sum_{i=1}^{n} v_i^t \left( o_i^{[m]} \right) - v_i^t \left( o_i^T \right).$$

Similarly, if $T = [m]$, it is not hard to see that we can write $R_T^{\vec{v}^t}$ as the intersection of $2^m - 1$ halfspaces.

We can also analyze the allocation of an MBARP parameterized by $(c, r_1, \ldots, r_m)$ without Bidder $i$'s participation for any $i \in [n]$, which we need to do in order to analyze the revenue function. To this end, for all $T \subseteq [m]$, let $\mathcal{O}_{T_{-i}}$ be the set of all allocations where exactly the elements of $T$ are allocated to all of the bidders except $i$, and let

$$\vec{o}^{T_{-i}} = \operatorname*{argmax}_{\vec{o} \in \mathcal{O}_{T_{-i}}} \left\{ \sum_{j \neq i} v_j^t \left( o_j \right) \right\}.$$

Again, regardless of the reserve prices, if $T$ consists of the items allocated by an MBARP without Bidder $i$'s participation, then $\vec{o}^{T_{-i}}$ will be the allocation. Now, for an MBARP parameterized by $(c, r_1, \ldots, r_m)$ without Bidder $i$'s participation, the allocation will be

$$\vec{o}^{T_{-i}} = \operatorname{argmax} \left\{ \sum_{j \neq i} v_j^t \left( o_j^{[m]_{-i}} \right) + c, \left\{ \sum_{j \neq i} v_j^t \left( o_j^{T_{-i}} \right) + \sum_{\ell \notin T} r_\ell \right\}_{T \neq [m]} \right\}.$$

For any $T \subseteq [m]$, let $R_{T_{-i}}^{\vec{v}^t}$ be the subset of $\mathbb{R}^{m+1}$ such that if an MBARP is parameterized by $(c, r_1, \ldots, r_m) \in R_{T_{-i}}^{\vec{v}^t}$, then the allocation of the MBARP without Bidder $i$'s partitipation on $\vec{v}$ is $\vec{o}^{T_{-i}}$. This means that if $T \neq [m]$, then just as before, $(c, r_1, \ldots, r_m) \in R_{T_{-i}}^{\vec{v}^t}$ if and only if it falls in the intersection of $2^m - 1$ halfspaces:

$$\sum_{\ell \notin T} r_\ell - \sum_{\ell \notin T'} r_\ell \geq \sum_{j \neq i} v_j \left( o_j^{T'_{-i}} \right) - \sum_{j \neq i} v_j \left( o_j^{T_{-i}} \right) \qquad \forall T' \notin \{T, [m]\}$$

$$\sum_{\ell \notin T} r_\ell - c \geq \sum_{j \neq i} v_j \left( o_j^{[m]_{-i}} \right) - \sum_{j \neq i} v_j \left( o_j^{T_{-i}} \right).$$

Similarly, if $T = [m]$, we can write $R_{T_{-i}}^{\vec{v}}$ as the intersection of $2^m - 1$ halfspaces.

Clearly, $\left\{ R_T^{\vec{v}^t} \right\}_{T \subseteq [m]}$ partition $\mathbb{R}^{m+1}$, since there will always be some allocation of an MBARP parameterized by an arbitrary point in $\mathbb{R}^{m+1}$. Similarly, $\left\{ R_{T_{-i}}^{\vec{v}^t} \right\}_{T \subseteq [m]}$ partition $\mathbb{R}^{m+1}$ for every $i \in [n]$.

Now, suppose

$$(c, r_1, \ldots, r_m) \in R_{T^0}^{\vec{v}^t} \bigcap R_{T^1_{-1}}^{\vec{v}^t} \bigcap \cdots \bigcap R_{T^n_{-n}}^{\vec{v}^t} = R$$

for some $T^0, T^1, \ldots, T^n \subseteq [m]$. We show that $rev_{\vec{v}^t}(c, r_1, \ldots, r_m)$ is linear on $R$ by splitting the analysis into four cases.

1. If $T^0, T^1, \ldots, T^n \neq [m]$ we can write

$$rev_{\vec{v}^t}(c, r_1, \ldots, r_m) = \sum_{i=1}^n \left[ \sum_{j \neq i} \left( v_j^t \left( o_j^{T^i_{-i}} \right) - v_j^t \left( o_j^{T^0} \right) \right) + \sum_{\ell \notin T^i} r_\ell - \sum_{\ell \notin T^0} r_\ell \right]. \quad (5)$$

2. If $T^0 \neq [m]$ and $T^i = [m]$ for some $i \in \{1, \ldots, n\}$, then we replace the summand of Equation 5 indexed by $i$ with

$$\sum_{j \neq i} \left( v_j^t \left( o_j^{T^i_{-i}} \right) - v_j^t \left( o_j^{T^0} \right) \right) + c - \sum_{\ell \notin T^0} r_\ell.$$

3. If $T^0 = [m]$ and $T_1, \ldots, T_n \neq [m]$, then

$$rev_{\vec{v}^t}(c, r_1, \ldots, r_m) = \sum_{i=1}^n \left[ \sum_{j \neq i} \left( v_j^t \left( o_j^{T^i_{-i}} \right) - v_j^t \left( o_j^{T^0} \right) \right) + \sum_{\ell \notin T^i} r_\ell - c \right]. \quad (6)$$

4. If $T^0 = [m]$ and $T^i = [m]$ for some $i \in \{1, \ldots, n\}$, then we replace the summand of Equation 6 indexed by $i$ with

$$\sum_{j \neq i} \left( v_j^t \left( o_j^{T^i_{-i}} \right) - v_j^t \left( o_j^{T^0} \right) \right).$$

In all of these cases, $rev_{\vec{v}^t}(c, r_1, \ldots, r_m)$ is a linear function over

$$(c, r_1, \ldots, r_m) \in R_{T^0}^{\vec{v}^t} \bigcap R_{T^1_{-1}}^{\vec{v}^t} \bigcap \cdots \bigcap R_{T^n_{-n}}^{\vec{v}^t} = R.$$

To summarize, we fixed $\vec{v}^t \in \mathcal{S}$ and introduced $n+1$ partitions of $\mathbb{R}^{m+1}$. Each partition is made up of $2^m$ cells and each cell is defined as the intersection of $2^m - 1$ halfspaces. If we restrict the domain of the revenue function to the intersection of any $n+1$ cells, one from each of the $n+1$ partitions, then the revenue function on that restricted domain is linear and therefore, there is one subregion where $rev_{\vec{v}^t}(c, r_1, \ldots, r_m)$ exceeds its target revenue and one subregion where it does not.

One generous upper bound on the number of different regions induced by taking the intersection of any $n+1$ cells, one from each of the $n+1$ partitions, is the number of different regions induced by the $(n+1)2^m(2^m-1)$ total hyperplanes. This is at most $(m+1)\left((n+1)2^m(2^m-1)\right)^{m+1} \leq (m+1)\left((n+1)4^m\right)^{m+1}$ because the number of regions induced by $k$ hyperplanes in $\mathbb{R}^d$ is at most $\sum_{i=1}^d \binom{k}{i} \leq dk^d$. Again, if we restrict the domain of the revenue function to any of these induced regions, the revenue function will be linear. As we saw in cases (1)-(4) of our case analysis, depending on the region, the revenue function will take a specific linear form. For a given region $R$, denote this specific linear form of the revenue function on by $R$ as $rev_{\vec{v}^j}|_R$. With this in mind, we define one more hyperplane per region: $rev_{\vec{v}^j}|_R > z^j$. In total, this contributes at most $(m+1)\left((n+1)4^m\right)^{m+1}$ more hyperplanes, since this is the maximum number of induced regions on $\mathbb{R}^{m+1}$. We are therefore left with at most $\alpha = (m+1)\left((n+1)4^m\right)^{m+1} + (n+1)2^m(2^m-1) = O\left(mn^m 8^{m^2}\right)$ total hyperplanes per valuation vector function $\vec{v}^j \in \mathcal{S}$.

If we merge all of the $N$ sets of $\alpha$ hyperplanes, $\mathbb{R}^{m+1}$ will be split into at most $(m+1)(N\alpha)^{m+1}$ regions, each of which induces a specific binary labeling of $\mathcal{S}$ (whether or not $\vec{v}^j$ exceeds its target revenue). Therefore, it must be that $2^N \leq (m+1)(N\alpha)^{m+1}$, so $N = O(m \log \alpha) = O\left(m^3 \log n\right)$. □

## C.1 Bundle Reserve Prices Lower Bound

In this section, we justify our choice to concentrate on MBARPs with item-specific reserve prices. In particular, we prove that no algorithm can learn over the class of $n$-bidder, $m$-item MBARP revenue functions with bundle-specific reserve prices using sample complexity $o\left(4^m/\sqrt{m}\right)$.

As in the proof of Theorem 8, we construct a special set $V$ of valuation functions. In this case, $V$ is a set of single-bidder, $m$-item valuation functions and $|V| = \Omega\left(4^m/\sqrt{m}\right)$. We then show that for any subset $H$ of $V$, there exists a setting of the bundle reserve prices that has high revenue over valuation functions in $H$, but low revenue on the valuation functions in $V \setminus H$. We describe $V$ more formally in Theorem 10. As we show in Remark 1, this construction can trivially be extended to a set of $n$-bidder, $m$-item valuation functions of the same size. Then, as in Theorem 8, this immediately implies hardness for learning over the uniform distribution on $V$. We provide the construction of $V$ and, given the parallel proof structure, we refer the reader to Theorem 8 to see how this implies hardness of learning.

We now present the construction of the set of valuation functions $V$.

**Theorem 10.** *For any $m \geq 2$, there exists a set of $N = \Omega\left(\frac{4^m}{\sqrt{m}}\right)$ single-bidder, $m$-item valuation function vectors $V = \left\{\vec{v}^1, \ldots, \vec{v}^N\right\}$ such that for any $H \subseteq V$, there exists a set of monotone bundle reserve prices such that the resulting auction has revenue 0 on $\vec{v}^i$ if $\vec{v}^i \in H$ and revenue $1 - \gamma$ on $\vec{v}^i$ if $\vec{v}^i \notin H$, for any $\gamma \in (0, 1)$.*

*Proof.* We define the set $V = \left\{\vec{v}^1, \ldots, \vec{v}^N\right\}$ of single-bidder valuation functions, where $\vec{v}^j = \left(v_1^j(b_1), \ldots, v_1^j(b_{2^m})\right)$. Assume for now that $m$ is even, and let $\tilde{b}_1, \ldots, \tilde{b}_N$ be a fixed ordering of the subsets of $2^{[m]}$ of size $m/2$, so $N = \Omega\left(\frac{4^m}{\sqrt{m}}\right)$. Let $\vec{v}^\ell$ for $\ell \in [N]$ be defined as follows.

$$
v_1^\ell(b_i) = \begin{cases} 0 & \text{if } |b_i| < m/2 \\ 1 & \text{if } b_i = \tilde{b}_\ell \\ 0 & \text{if } |b_i| = m/2 \text{ and } b_i \neq \tilde{b}_\ell \\ 1 & \text{if } |b_i| > m/2 \end{cases}.
$$

We claim that for any $H \subseteq V$, there exists a set of monotone bundle reserve prices $\{r(b_1), \ldots, r(b_{2^m})\}$ such that the resulting auction has 0 revenue on all valuation functions $\vec{v}^i$ such that $\vec{v}^i \notin H$ and $1 - \gamma$ revenue on all valuation functions $\vec{v}^i \in H$. The reserve prices are defined as follows:

$$
v_0(b_i) = r(b_i) = \begin{cases} 0 & \text{if } |b_i| < m/2 \\ 1 - \gamma & \text{if } b_i = \tilde{b}_\ell^c \text{ and } \vec{v}^\ell \notin H \\ 0 & \text{if } b_i = \tilde{b}_\ell^c \text{ and } \vec{v}^\ell \in H \\ 1 - \gamma & |b_i| > m/2 \end{cases}.
$$

Regardless of whether or not $\vec{v}^\ell$ is in $H$, $\left(\tilde{b}_\ell, \tilde{b}_\ell^c\right) = \text{argmax}_{\vec{\sigma} \in \mathcal{O}}\left\{v_0(o_0) + v_1^\ell(o_1)\right\}$ and $1 - \gamma = \max_{b_i \in 2^{[m]}}\left\{v_0(b_i)\right\}$. Therefore, Bidder 1 pays

$$
1 - \gamma - v_0\left(\tilde{b}_\ell^c\right) = \begin{cases} 1 - \gamma & \text{if } \vec{v}^\ell \in H \\ 0 & \text{if } \vec{v}^\ell \notin H \end{cases}.
$$

□

**Remark 1.** *For any $m \geq 2, n \geq 1$, there exists a set of $N = \Omega\left(4^m/\sqrt{m}\right)$ $n$-bidder valuation function vectors $V = \left\{\vec{v}^1, \ldots, \vec{v}^N\right\}$ such that for any $H \subseteq V$, there exists a set of monotone bundle reserve prices such that the resulting auction has revenue 0 on $\vec{v}^i$ if $\vec{v}^i \notin H$ and revenue $1 - \epsilon$ on $\vec{v}^i$ if $\vec{v}^i \in H$.*

This follows simply by setting $v_1^\ell$ as in the proof of Theorem 10 and setting $v_j^\ell(b_i) = 0$ for all $\ell \in [N], i \in [2^m], j \neq 1$.

## D  Sample complexity bounds for approximation algorithms

**Theorem 11.** *Let $\mathcal{S} = \{x_1, \ldots, x_N\}$ be a sample drawn from $\mathcal{D}$ and $\epsilon, \delta \in (0,1)$ be given. Suppose that $N$ is sufficiently large to ensure that with probability at least $1 - \delta/2$, for any $h \in \mathcal{H}$, $\mathbb{E}_{x \sim \mathcal{D}}[\ell(h,x)] - \frac{1}{N} \sum_{i=1}^N \ell(h, x_i) < \epsilon$.*

*Suppose $h^* \in \mathcal{H}$ is a function that minimizes expected loss with respect to the distribution, $\hat{h}$ is a function that minimizes average loss over the sample $\mathcal{S}$, and $\tilde{h} \in \mathcal{H}$ is a function such that the average loss of $\hat{h}$ over $\mathcal{S}$ is within an additive $\rho$ factor of the average loss of $\tilde{h}$ over $\mathcal{S}$. In other words, $\frac{1}{N} \sum_{i=1}^N \ell\left(\tilde{h}, x_i\right) - \frac{1}{N} \sum_{i=1}^N \ell\left(\hat{h}, x_i\right) \leq \rho$ for some $\rho > 0$. Then with probability at least $1 - \delta$,*

$$\mathbb{E}_{x \sim \mathcal{D}}\left[\ell\left(\tilde{h}, x\right)\right] - \mathbb{E}_{x \sim \mathcal{D}}[\ell(h^*, x)] \leq \epsilon + c\sqrt{\frac{\ln(4/\delta)}{2N}} + \rho.$$

*Meanwhile, if $\frac{1}{N} \sum_{i=1}^N \ell\left(\tilde{h}, x_i\right) \leq (1+\alpha)\frac{1}{N} \sum_{i=1}^N \ell\left(\hat{h}, x_i\right)$ for some $\alpha \in [0,1)$, then*

$$\mathbb{E}_{x \sim \mathcal{D}}\left[\ell\left(\tilde{h}, x\right)\right] - \mathbb{E}_{x \sim \mathcal{D}}[\ell(h^*, x)] \leq \epsilon + (1+\alpha)\left(c\sqrt{\frac{\ln(4/\delta)}{2N}}\right) + \alpha \mathbb{E}_{x \sim \mathcal{D}}[\ell(h^*, x)].$$

*Moreover, both bounds are tight in the worst case.*

*Proof.* First, let $\epsilon = 2R_N(\mathcal{H}) + c\sqrt{\frac{2\ln(4/\delta)}{N}}$. For ease of notation, for any $h \in \mathcal{H}$, let $L_\mathcal{S}(h) = \frac{1}{N} \sum_{i=1}^N \ell(h, x_i)$ and $L_\mathcal{D}(h) = \mathbb{E}_{x \sim \mathcal{D}}[\ell(h, x)]$. Suppose that $h^*$ is the optimal hypothesis in $\mathcal{H}$ (i.e. it minimizes $L_\mathcal{D}(h)$, the expected loss over the distribution $\mathcal{D}$), $\hat{h}$ is the empirical risk minimizer (i.e. it minimizes $L_\mathcal{S}(h)$, the average loss over the sample $\mathcal{S}$), and $\tilde{h}$ is a hypothesis such that $L_\mathcal{S}\left(\tilde{h}\right) - L_\mathcal{S}\left(\hat{h}\right) \leq \rho$ for some $\rho > 0$. Then with probability at least $1 - \delta$,

$$L_\mathcal{D}\left(\tilde{h}\right) - \epsilon \leq L_\mathcal{S}\left(\tilde{h}\right) \tag{7}$$

$$\leq L_\mathcal{S}\left(\hat{h}\right) + \rho \tag{8}$$

$$\leq L_\mathcal{S}(h^*) + \rho \tag{9}$$

$$\leq L_\mathcal{D}(h^*) + c\sqrt{\frac{2\ln(4/\delta)}{2N}} + \rho. \tag{10}$$

Inequality 7 follows from Equation standard Rademacher complexity uniform convergence bounds: with probability at least $1 - \delta/2$, $L_\mathcal{D}\left(\tilde{h}\right) - L_\mathcal{S}\left(\tilde{h}\right) \leq \epsilon$ (see, for example, Shalev-Shwartz and Ben-David [2014]). Inequality 8 follows from the fact that $L_\mathcal{S}\left(\tilde{h}\right) \leq L_\mathcal{S}\left(\hat{h}\right) + \rho$. Inequality 9 follows because $\hat{h}$ is the empirical risk minimizer (i.e. it minimizes $L_\mathcal{S}(h)$). Finally, inequality 10 is a result, again, of Hoeffding's inequality, which guarantees that with probability at least $1 - \delta/2$, $L_\mathcal{S}(h^*) \leq L_\mathcal{D}(h^*) + c\sqrt{\frac{2\ln(4/\delta)}{2N}}$.

Rearranging, we get that

$$L_\mathcal{D}\left(\tilde{h}\right) - L_\mathcal{D}(h^*) \leq \epsilon + c\sqrt{\frac{2\ln(4/\delta)}{2N}} + \rho,$$

as claimed.

Next, suppose that $\tilde{h}$ is a hypothesis such that $L_{\mathcal{S}}\left(\tilde{h}\right) \leq (1+\alpha)L_{\mathcal{S}}\left(\hat{h}\right)$. We similarly can deduce that with probability at least $1 - \delta$,

$$L_{\mathcal{D}}\left(\tilde{h}\right) - \epsilon \leq L_{\mathcal{S}}\left(\tilde{h}\right) \tag{11}$$

$$\leq (1+\alpha)L_{\mathcal{S}}\left(\hat{h}\right) \tag{12}$$

$$\leq (1+\alpha)L_{\mathcal{S}}\left(h^*\right) \tag{13}$$

$$\leq (1+\alpha)\left(L_{\mathcal{D}}\left(h^*\right) + c\sqrt{\frac{2\ln(4/\delta)}{2N}}\right). \tag{14}$$

Rearranging, we get that

$$L_{\mathcal{D}}\left(\tilde{h}\right) \leq \epsilon + (1+\alpha)L_{\mathcal{D}}\left(h^*\right) + (1+\alpha)c\sqrt{\frac{2\ln(4/\delta)}{2N}},$$

which means that

$$L_{\mathcal{D}}\left(\tilde{h}\right) - L_{\mathcal{D}}\left(h^*\right) \leq \epsilon + (1+\alpha)c\sqrt{\frac{2\ln(4/\delta)}{2N}} + \alpha L_{\mathcal{D}}\left(h^*\right),$$

as desired.

We remark that inequalities 7-14 could be tight in the worst case, so both bounds are tight.

$\square$

## Footnotes

[1]A mechanism is *ex post* strategy-proof if truthful bidding is an *ex post Nash equilibrium* in which all bidders always receive nonnegative utility. By *ex post Nash equilibrium*, we mean that for each player, no matter the valuations of the other players but given that they are bidding truthfully, that player will maximize her utility if she bids truthfully as well.

[2]A mechanism is an *almost affine maximizer* if it is an affine maximizer for sufficiently high valuations. Lavi et al. [2003] conjecture that the "almost" qualifier is merely technical, and can be removed in future research.