[Reviews · NeurIPS 2016]

Reviewer 1

Summary

This paper deals with the sample complexity of automated mechanism design for the problem of maximizing the revenue in a combinatorial auction (CA). Given a class of auction mechanisms, the automated mechanism design takes as input samples from the bidders’ valuation distribution (which, in practice, may be the history records from the previous auctions), and output the choice of auction mechanism with high revenue. This work presents several upper bounds on the sample complexities for various auction classes. Although Morgenstern and Roughgarden (reference [19] in this paper) studied the same problem of bounding the sample complexities of CA, their work only deals with “simple auctions” which can be reduced to the single-bidder setting. In contrast, this paper studies the hierarchy of deterministic CA families consists of VCG-based mechanisms. As opposed to sequential auctions, these auction classes have more complex structure, and provide the auctioneer more degrees of freedom in designing mechanism. In particular, the authors study the affine maximizer auction (AMA), the virtual valuation CA (VVCA), the lambda-auction, the mixed bundling auction with reserve prices (MBARPs) and the mixed bundling auction (MBA). For AMA, VVCA and lambda-auction, the authors provide certain negative results. Firstly, an exponential upper bound on the Rademacher complexity of the revenue function is given for the class AMA. (Theorem 2) Based on this, an exponential upper bound on sample complexity for all these three classes is provided, while two exponential lower bounds are provided for lambda-auction and VVCA respectively as complementary results (Theorem 1). For the less general models MBARPs and MBA, on the other hand, many positive results are given. For MBA with only 2 bidders, the authors first show that the pseudo-dimension is 2 (theorem 3), and provide a polynomial upper bound on sample complexity (Theorem 4). Then the authors generalize these results for the general case with n bidders (Theorem 5, 6). Finally, the authors provide polynomial upper bounds on the pseudo-dimension and sample complexity of MBARPs.

Qualitative Assessment

This paper appears to be mostly a followup on the work of [19], "learning simple auctions" where techniques used to prove generalization error bounds were used to help reason about the design of mechanisms. In the present submission, the authors focus a much broader classes of auctions (affine maximize auction, virtual valuation CA, lambda-auction, mixed bundling auction with reserve price, and mixed bundling auction), and provide much more general results. The techniques and results are interesting. But in the context of the previous work already introducing the underlying idea that you can apply learning theory to mechanism design problems of this form, I do wonder whether the results are suitably exciting to deserve publication in NIPS. (The paper is clearly above the bar for other conference venues)

Confidence in this Review

2-Confident (read it all; understood it all reasonably well)


Reviewer 2

Summary

This paper is a nice application of learning theory to mechanism design for combinatorial auctions. In particular, given a class of strategy-proof mechanisms, the paper answers the question of how well the revenue-optimal mechanism on a set of bids/values sampled from a distribution generalizes to the true distribution. The authors provide results for at five general classes of mechanisms (as well as a result for approximations). For the most general classes (affine maximizer auctions, virtual valuations, lambda-auctions), they state exponential lower bounds for uniform convergence. However, they are able to show that mixed bundling auctions have polylogarithmic pseudo-dimension in the number of bidders and items.

Qualitative Assessment

Given the importance of combinatorial auctions, it is surprising that few sample complexity results exist in the literature. Notably, it appears that the recent work of Morgenstern and Roughgarden does not apply to the settings considered here. Although I did not check the proofs in full, the main ideas behind the pseudo-dimension constructions are well-presented and informative. Overall, I think the problem studied is important and potentially very impactful.

Confidence in this Review

2-Confident (read it all; understood it all reasonably well)


Reviewer 3

Summary

The authors analyze the sample complexity of learning approximately optimal auctions from among well-studied combinatorial auction classes. The authors analyze what is the relation of the best in sample mechanism in the class with the revenue of this mechanism on the actual distribution of player's values. The authors heavily use the toolbox of Rademacher complexities and Rademacher calculus, combined with more combinatorial approaches, such as the the VC dimension, in order to bound the Rademacher complexity of the revenue objective from among well-studied classes of combinatorial auctions.

Qualitative Assessment

The results are technically interesting both from a learning theoretic point of view, as they analyze quite complex functions classes and manage to bound their complexity, as well as from a mechanism design point of view, as they introduce the Rademacher complexity machinery to mechanism desing. For this reason, I recommend acceptance.

Confidence in this Review

2-Confident (read it all; understood it all reasonably well)


Reviewer 4

Summary

Considers sample complexity of automated mechanism design for revenue: Given a bunch of samples of bidders' valuations over bundles of items, choose an allocation and payment rule to be used on a new sample of bidders, such that expected revenue is approximately maximized. Considers classes of VCG-like mechanisms which I believe are widely used/considered. This boils the problem down to selecting weights on each bidder and allocation (an exponentially large space).

Qualitative Assessment

I like the paper and results. I was not sure how to interpret the bounds sometimes, as they can be complicated or involve, for instance, the H constants which I am not sure how to interpret. So a bit of guidance would help, for instance, if the main takeaway of Section 3.1 is "it's exponential", maybe say that directly. But overall the message is clear. One question I am left with is how to understand the role of uniform convergence in the negative results. How strong of a negative result is it to say that convergence over all auctions in the class requires so many samples? A bit more discussion about this, and obstacles to some stronger goal (e.g. sample complexity of finding an approximately optimal auction), would be interesting.

Confidence in this Review

1-Less confident (might not have understood significant parts)


Reviewer 5

Summary

The paper provides the sample complexity analysis of automated mechanism design which is used for searching of high-revenue combinatorial auction in an auction class. The main results are uniform convergence bounds of revenue for a hierarchical of auction classes, which state that exponential sample complexity are not avoidable for general AMA auctions, VVCA auctions and \lambda auctions, meanwhile polynomial sample complexity can be achieved for MBARP and MBA auctions. The core technique involved is bounding the complexity of different auction classes, with Rademacher complexity or pseudo dimension.

Qualitative Assessment

+ This paper explores an interesting new research area for learning theory. Based on traditional technique for sample complexity analysis, the results reveals new insights on the structure of auction class hierarchy. The paper is well-written so that the content is easy to follow. - The quantity "U" appeared in bounds are not explained in the paper.

Confidence in this Review

2-Confident (read it all; understood it all reasonably well)